



# Assessing the robustness and scalability of the accelerated pseudo-transient method towards exascale computing

Ludovic Räss[1,2,*], Ivan Utkin[1,3,*], Thibault Duretz[4,5], Samuel Omlin[6], and Yuri Y. Podladchikov[7,8]

[1]Laboratory of Hydraulics, Hydrology and Glaciology (VAW), ETH Zurich, Zurich, Switzerland
[2]Swiss Federal Institute for Forest, Snow and Landscape Research (WSL), Birmensdorf, Switzerland
[3]Faculty of Mechanics and Mathematics, Lomonosov Moscow State University, Moscow, Russia
[4]Institut für Geowissenschaften, Geothe-Universität Frankfurt, Frankfurt, Germany
[5]Univ. Rennes, CNRS, Géosciences Rennes UMR 6118, F-35000 Rennes, France
[6]Swiss National Supercomputing Centre (CSCS), ETH Zurich, Lugano, Switzerland
[7]Institute of Earth sciences, University of Lausanne, Lausanne, Switzerland
[8]Swiss Geocomputing Centre, University of Lausanne, Lausanne, Switzerland

**Correspondence:** Ludovic Räss (ludovic.rass@gmail.com)

**Abstract.** The development of highly efficient, robust and scalable numerical algorithms lags behind the rapid increase in massive parallelism of modern hardware. We address this challenge with the accelerated pseudo-transient iterative method and present here a physically motivated derivation. We analytically determine optimal iteration parameters for a variety of basic physical processes and confirm the validity of theoretical predictions with numerical experiments. We provide an efficient
numerical implementation of pseudo-transient solvers on graphical processing units (GPUs) using the Julia language. We achieve a parallel efficiency over 96% on 2197 GPUs in distributed memory parallelisation weak scaling benchmarks. 2197 GPUs allow for unprecedented terascale solutions of 3D variable viscosity Stokes flow on $4995^3$ grid cells involving over 1.2 trillion degrees of freedom. We verify the robustness of the method by handling contrasts up to 9 orders of magnitude in material parameters such as viscosity, and arbitrary distribution of viscous inclusions for different flow configurations.
Moreover, we show that this method is well suited to tackle strongly nonlinear problems such as shear-banding in a visco-elasto-plastic medium. A GPU-based implementation can outperform CPU-based direct-iterative solvers in terms of wall-time even at relatively low resolution. We additionally motivate the accessibility of the method by its conciseness, flexibility, physically motivated derivation and ease of implementation. This solution strategy has thus a great potential for future high-performance computing applications, and for paving the road to exascale in the geosciences and beyond.

## 1 Introduction

The recent development of multi-core devices has lead to the democratisation of parallel computing. Since the "memory-wall" in the early 2000's (Wulf and McKee, 1995), the continuous increase of the ratio between computation speed and memory access speed results in a shift from compute-bound to memory-bound algorithms. Currently, multi-core devices such as graphical processing units (GPUs) feature floating point arithmetic processing rates outperforming memory access rates by over an





order of magnitude. As a result, memory accesses constitute the main performance limiter in a majority of scientific computing applications, arithmetic efficiency becoming much less performance relevant.

The current computing landscape challenges scientific computing applications seeking at solutions to partial differential equations (PDEs) and their legacy implementations that rely on non-local methods, one example being matrix factorisation-based solvers. The main reasons these applications do not perform optimally on modern hardware are their prohibitive and

nonlinear memory utilisation increase as function of numbers of degrees of freedom (DoFs), proportional to the global problem size or the spatial numerical resolution. As a result, the usage of sparse direct solvers in high-performance computing (HPC) is only possible for relatively small-scale problems due to excessive memory and computational resources requirements, inherently responsible for limitations in parallel scalability. Even storing the sparse matrix structure and nonzero elements in a compressed form is often not possible due to the limited amount of available memory. This situation naturally increases the

attractivity of iterative matrix-free algorithms for solving large-scale problems.

Pseudo-transient (PT), or dynamic relaxation (DR) methods see a regain in active development over the last decades. Pseudo-transient methods are matrix-free and construct upon a transient physics analogy to establish a stationary solution. Unlike Krylov-type methods such as the conjugate gradient method, PT methods build upon a fixed-point iteration, in which the update of each grid point is entirely local and does not require global reductions (and thus global communication) at each step

of the algorithm. Due to the locality of the algorithm, software implementations can achieve very high per-node performance and near-ideal scaling on distributed memory systems with accelerators such as GPUs.

PT methods build upon a physical description of a process. It becomes therefore possible to model strongly nonlinear processes and achieve convergence starting from nearly arbitrary initial conditions. Conventional linearisation methods such as the Newton-Raphson method may fail to converge if the initial approximation is not close enough to the solution. Examples are

problems of resolving strain localisation owing to plastic yielding (Duretz et al., 2019a) or non-Newtonian power-law rheology, as well as nonlinearities arising from multi-physics coupling such as shear heating (Duretz et al., 2019b; Räss et al., 2020) and two-phase flow localisation (Räss et al., 2018; Räss et al., 2019a).

The implementation conciseness constitutes another advantage of PT methods compared to matrix-based solvers. Numerical codes implementing PT algorithms characterise as concise and short, the explicit pseudo-time integration preserving the

similarity to the mathematical description of the system of PDEs. Conciseness supports efficient and thus faster development and significantly simplifies the addition of new physics, a crucial step when investigating multi-physics couplings. Also, the similarity between mathematical and discretised code notation makes PT methods an attractive tool for research and education.

The PT method originated as a "dynamic relaxation" method in the 1960s, when it was, e.g., applied for calculating the stresses and displacements in concrete pressure vessels (Otter, 1965). They build upon pioneering work by Richardson (1911)

proposing an iterative solution approach to PDEs related to dam-engineering calculations. In geosciences, the PT method was introduced by Cundall (1976) as the FLAC (Fast Lagrangian Analysis of Continua) algorithm. Subsequently, the FLAC method was successfully applied to simulate the Rayleigh-Taylor instability in visco-elastic flow (Poliakov et al., 1993), as well as the formation of shear bands in rocks (Poliakov et al., 1994). Among other applications of the PT method are structural analysis problems including failure (Kilic and Madenci, 2009), buckling (Ramesh and Krishnamoorthy, 1993), and form-





finding (Barnes, 1999). The DR terminology is still reference in the finite-element method (FEM) community (Rezaiee-Pajand et al., 2011).

Interestingly, Richardson developed his iterative approach without being aware of the work of Gauss and Seidel, their method being named the Liebmann method when applied to solving PDEs. Early development of iterative algorithms such as one-dimensional projection methods and Richardson iterations depend on the current iterate only; they were well-suited for
early low memory computers, however lacking in efficient convergence rates. The situation changed in 1950, when Frankel introduced second-order iterations as extension of the Richardson and Liebmann methods, adding dependency on the previous iterate (Frankel, 1950), resulting in the second-order Richardson and extrapolated Liebmann methods, respectively. These methods feature enhanced convergence rates (Young, 1972), and perform on par, the first being slightly more interesting as fully local (Riley, 1954). By analogy with the explicit solution to time-dependent PDEs, Frankel introduced additional "physically
motivated" terms in his iterative scheme. Since the Chebyshev iteration can be recovered for constant parameters, second-order or extrapolated methods are also termed semi-iterative. Note that one challenge related to Chebyshev semi-iterative methods relies in the need of an accurate estimate of extremal eigenvalues relating to the interval in which the residual is minimised. The review by Saad (2020) provides further interesting development insights.

The accelerated PT method for elliptic equations is mathematically equivalent to the second-order Richardson rule (Frankel,
1950; Riley, 1954; Otter et al., 1966). The convergence rate of PT methods is very sensitive to the iteration parameters' choice. For the simplest problems, e.g., the stationary heat conduction in a rectangular domain described by the Laplace's equation, these parameters can be derived analytically based on the analysis of the damped wave equation (Cox and Zuazua, 1994). In the general case, the values of these parameters are associated with the maximum eigenvalue of the stiffness matrix. The eigenvalue problem is computationally intensive, and for practical purposes the eigenvalues are often approximated based on
the Rayleigh's quotient or Gershgorin's theorem (Papadrakakis, 1981). Thus, the effective application of PT methods relies on an efficient method to determine the iteration parameters. In the last decades, several improvements were made to the stability and convergence rate of DR methods (Cassell and Hobbs, 1976; Rezaiee-Pajand et al., 2011; Alamatian, 2012). Determining the general and efficient procedure for estimating the iteration parameters still remains an active area of research.

We identify among current challenges for iterative methods three important ones, namely (1) ensure the iteration count to
scale linearly with numerical resolution increase, possibly independently of material parameters contrasts and nonlinearities, (2) achieve minimal per-device main memory access redundancy at maximal access speed, and (3) achieve a parallel efficiency close to 100% on multi-device –distributed memory– systems. In this study, we address (1) by presenting the accelerated PT method, resolving several types of basic physical processes. We consider (2) and (3) as challenges partly related to scientific software design and engineering; we address them using the emerging Julia language (Bezanson et al., 2017), which solves the
"two-language problem" and provides the missing tool towards making prototype and production code become one and breaking up the technically imposed hard division of the software stack into domain science tasks (higher levels of the stack) and computer science tasks (lower levels of the stack). The Julia applications featured in this study rely on recent Julia package developments undertaken by the authors to empower domain scientists to write architecture-agnostic high-level code for parallel high-performance stencil computations on massively parallel hardware such as latest GPU-accelerated supercomputers.





In this work, we present the results of analytical analysis of the pseudo-transient equations for (non-)linear diffusion and incompressible visco-elastic Stokes flow problems. We motivate our selection of particular physical processes as a broad range of natural processes categorise mathematically either as diffusive, wave-like or mechanical processes and thus constitute the main building blocks of multi-physics applications. We derive iteration parameters' approximations from continuous, non-discretised formulations with emphasis on analogy between these parameters and non-dimensional numbers arising in mathematical modelling of physical processes. Such a physics-inspired numerical optimisation approach has the advantage to provide a framework building upon solid classical knowledge and for which various analytical approaches exist to derive or optimise parameters of interest. We assess the algorithmic and implementation performance and scalability of the 2D and 3D numerical Julia (multi-)GPU (non-)linear diffusion and visco-elastic Stokes flow implementations. We report scalability beyond terrascale number of DoFs on up to 2197 Nvidia Tesla P100 GPUs on the *Piz Daint* supercomputer at the Swiss National Supercomputing Centre, CSCS. We demonstrate the versatility and the robustness of our approach in handling nonlinear problems by applying the accelerated pseudo-transient method to resolve spontaneous strain localisation in elasto-viscoplastic media in 2D and 3D, and comparing time to solution with direct-sparse solvers in 2D. We further demonstrate the convergence of the method to be mostly insensitive to arbitrary distributions of viscous inclusions with viscosity contrasts up to 9 orders of magnitude in the incompressible viscous Stokes flow limit.

The latest versions of the open-source Julia codes used in this study are available from GitHub within the PTsolvers organisation at https://github.com/PTsolvers/PseudoTransientDiffusion.jl (last access: 9 December 2021) and https://github.com/PTsolvers/PseudoTransientStokes.jl (last access: 9 December 2021). Past and future versions of the software are available from a permanent DOI repository (Zenodo) at: https://doi.org/10.5281/zenodo.5764691 (Räss and Utkin, 2021a) and https://doi.org/10.5281/zenodo.5764696 (Räss and Utkin, 2021b). The `README` files provide the instructions to get started reproducing majority of the presented results.

## 2 The pseudo-transient method

At the core of the pseudo-transient method lies the idea of considering stationary processes, often described by elliptic PDEs, as the limit of some transient processes described by parabolic or hyperbolic PDEs.

Pseudo-transient methods were present in literature since 1950's (Frankel, 1950) and have a long history. However, the equations describing processes under consideration are usually analysed in discretised form with little physical motivation. We here provide examples of pseudo-transient iterative strategies relying on physical processes as a starting point both for diffusion and incompressible visco-elastic Stokes problems. We further discuss how the choice of transient physical processes influences the performance of iterative methods and how to select optimal iteration parameters upon analysing the equations in their continuous form.

In the following, we make two assumptions:

1. The computational domain is a cube $x_k \in [0; L], k = 1 \dots n_d$, where $n_d$ is the number of spatial dimensions;





2. This domain is discretised with a uniform grid of cells. The number of grid cells is the same in each spatial dimension and is equal to $n_x$.

However, in practice, this solution strategy is not restricted to cubic meshes with similar resolution in each dimension.

## 2.1 Diffusion

Let us first consider the diffusion process:

$$\rho \frac{\partial H}{\partial t} = -\nabla_k q_k \tag{1}$$

$$q_i = -D\nabla_i H \tag{2}$$

where $H$ is some quantity, $D$ is the diffusion coefficient, $\rho$ is a proportionality coefficient and $t$ is the physical time.

By substituting Eq. (2) into (1) we obtain an equation for $H$:

$$\rho \frac{\partial H}{\partial t} = \nabla_k(D\nabla_k H) \tag{3}$$

which in case of $D = \text{const}$ is, e.g., the standard parabolic heat equation. Equation (3) must be supplemented with initial conditions at $t = 0$ and two boundary conditions for each spatial dimension at $x_k = 0$ and $x_k = L$. We here assume that Dirichlet boundary conditions are specified. The choice of the boundary conditions type affects only the values of optimal iteration parameters and does not limit the generality of the method.

Firstly, we consider a stationary diffusion process, which is described by Eq. (3) with $\partial H/\partial t \to 0$:

$$\nabla_k(D\nabla_k H) = 0 \tag{4}$$

Solving Eq. (4) numerically using conventional numerical methods would require to assemble a coefficient matrix and to rely on a direct or iterative sparse solver. Such approach may be preferred for 1D and some 2D problems, but since our aim is large-scale 3D modelling, we are interested in matrix-free iterative methods. In the following, we describe two of such methods, both of which are based on transient physics.

### 2.1.1 The first-order pseudo-transient method

Solution to Eq. (4) is achieved as a limit of the solution to the transient Eq. (3) at $t \to \infty$. Therefore, the natural iteration strategy is to integrate the system numerically in time until convergence, i.e., until changes in $H$, defined in some metric, are smaller then a predefined tolerance.

The simplest pseudo-transient method is to replace physical time $t$ in Eq. (15) by numerical pseudo-time $\tau$, and the physical parameter $\rho$ with a numerical parameter $\tilde{\rho}$:

$$\tilde{\rho} \frac{\partial H}{\partial \tau} = \nabla_k(D\nabla_k H) \tag{5}$$





We refer to $\tau$ as the "pseudo-time" because we are not interested in the distributions of $H$ at particular values of $\tau$, therefore, $\tau$ is relevant only for numerical purposes. The numerical parameter $\tilde{\rho}$ can be chosen arbitrarily.

The number of iterations, i.e., the number of steps in pseudo-time required to reach convergence of the simplest method described by Eq. (5), is proportional to $n_x^2$ (see Sect. A1 in the Appendix). Quadratic scaling makes the use of the simplest pseudo-transient method impractical for large problems.

One possible solution to circumvent the poor scaling properties of this first-order method would be to employ an unconditionally stable pseudo-time integration scheme. However, that would require solving systems of linear equations, making the solution cost of one iteration equal to the cost of solving the original steady-state problem. We are thus interested in a method that is not significantly more computationally expensive than the first-order scheme but that offers an improved scalability.

### 2.1.2 The accelerated pseudo-transient method

One of the known extensions to the classical model of diffusion incorporates inertial terms in the flux definition (Chester, 1963). That addition makes it possible to describe wave propagation in otherwise diffusive processes. Those inertial terms are usually neglected because the time of wave propagation and relaxation is small compared to the characteristic time of the process (Maxwell, 1867). The modified definition of the diffusive flux, originally derived by Maxwell from kinetic theory of ideal gas, takes the following form:

$$\theta_{\mathrm{r}}\frac{\partial q_i}{\partial \tau} + q_i = -D\nabla_i H \tag{6}$$

where $\theta_{\mathrm{r}}$ is the relaxation time.

A notable difference between the flux definition from Eq. (6) and (2) is that the resulting system type switches from parabolic to hyperbolic and describes not only diffusion, but wave propagation phenomena as well. Combining Eq. (1), replacing $t$ with $\tau$, and (6) to eliminate $q$ yields

$$\tilde{\rho}\theta_{\mathrm{r}}\frac{\partial^2 H}{\partial \tau^2} + \tilde{\rho}\frac{\partial H}{\partial \tau} = \nabla_k(D\nabla_k H) \,, \tag{7}$$

which for $D = \mathrm{const}$ is a damped wave equation that frequently occurs in various branches of physics (Pascal, 1986; Jordan and Puri, 1999). Contrary to the parabolic Eq. (5), the information signal in the damped wave equation propagates at finite speed $V_{\mathrm{p}} = \sqrt{D/\tilde{\rho}/\theta_{\mathrm{r}}}$.

The Eq. (7) includes two numerical parameters, $\tilde{\rho}$ and $\theta_{\mathrm{r}}$. The choice of these parameters significantly influences the performance and the stability of the pseudo-transient method. Converting the Eq. (7) to non-dimensional form allows to reduce the number of free parameters to only one non-dimensional quantity $\mathrm{Re} = \tilde{\rho}V_p L/D$, which can be interpreted as a Reynolds number.

Another restriction on the values of iteration parameters arises from the conditions for the numerical stability of the explicit time integration. The numerical pseudo-time step $\Delta\tau$ is related to the wave speed $V_p$ via the following stability condition:

$$\Delta\tau \leq \frac{C}{V_p}\Delta x \tag{8}$$





where $\Delta x = L/n_x$ is the spatial grid step and $C$ is a non-dimensional number determined for the linearised problem using a von Neumann stability analysis procedure. For the damped wave equation, Eq. (7), here considered, $C \approx 1/\sqrt{n_\mathrm{d}}$, where $n_\mathrm{d}$ is the number of spatial dimensions (Alkhimenkov et al., 2021a).

We choose parameters $\tilde{\rho}$ and $\theta_\mathrm{r}$ so that the stability condition (Eq. (8)) is satisfied for an arbitrary $\Delta\tau$. We introduce the numerical velocity $\widetilde{V} = \widetilde{C}\Delta x/\Delta\tau$, where $\widetilde{C} \leq C$ is an empirically determined parameter. We conclude from numerical experiments that using $\widetilde{C} \approx 0.95C$ is usually sufficient for stable convergence, however, for significantly nonlinear problems, lower values of $\widetilde{C}$ may be specified. Expressions for $\tilde{\rho}$ and $\theta_\mathrm{r}$ are obtained by taking into account the definition of Re and solving for $V_p = \widetilde{V}$:

$$\tilde{\rho} = \mathrm{Re}\frac{D}{\widetilde{V}L} \tag{9}$$

$$\theta_\mathrm{r} = \frac{D}{\tilde{\rho}\widetilde{V}^2} = \frac{L}{\mathrm{Re}\widetilde{V}} \tag{10}$$

Depending on the value of the parameter Re, the pseudo-transient process described by the damped wave equation (Eq. (7)) will be more or less diffusive. In case of $\mathrm{Re} \to \infty$, diffusion dominates, resulting in the accelerated pseudo-transient method to be equivalent to the first-order method described in the Sect. 2.1.1, regaining the non-desired quadratic scaling of the convergence rate. If, instead, $\mathrm{Re} \to 0$, the system is equivalent to the undamped wave equation, resulting in a never converging method, because waves do not attenuate. An optimal value of Re exists between these two limits, which leads to the fastest convergence.

To estimate the optimal value of Re, we analyse the spectral properties of Eq. (7). The solution to the damped wave equation is decomposed into a superposition of plane waves with particular amplitude, frequency, and decay rate. Substituting a plane wave solution into the equation yields the dispersion relation connecting the decay rate of the wave to its frequency and values of Re. Considering the solutions to this dispersion relation, it is possible to determine the optimal value of Re, denoted here as $\mathrm{Re}_\mathrm{opt}$. For near-optimal values of Re, the number of iterations required for the method to converge exhibits linear instead of quadratic dependence on the numerical grid resolution $n_x$, which is a substantial improvement compared to the first-order pseudo-transient method.

We present detailed explanations and derivations of the dispersion analysis of different problems in the Appendix A, leading to the optimal value of Re:

$$\mathrm{Re}_\mathrm{opt} = 2\pi \tag{11}$$

We quantify the convergence rate by the number of iterations $n_\mathrm{iter}$ required to reduce the maximal deviation of the solution to the pseudo-transient equation from the true solution to the corresponding stationary problem by a factor of $e$ (base of natural logarithm), divided by the number of grid cells $n_x$. Results of the dispersion analysis for the 1D stationary diffusion problem show $n_\mathrm{iter} \approx 0.3n_x$ given optimal values of Re (Fig. 1a). We estimate that reduction of the residual by 14 orders of magnitude requires only $\sim 10n_x$ iterations.

For simplicity we only consider the case $D = \mathrm{const}$ in the dispersion analysis. In that case, both $\tilde{\rho}$ and $\theta_\mathrm{r}$ are constant. If the physical properties vary in space, i.e., $D = D(x_k)$, the iteration parameters are no longer constant and must be locally defined





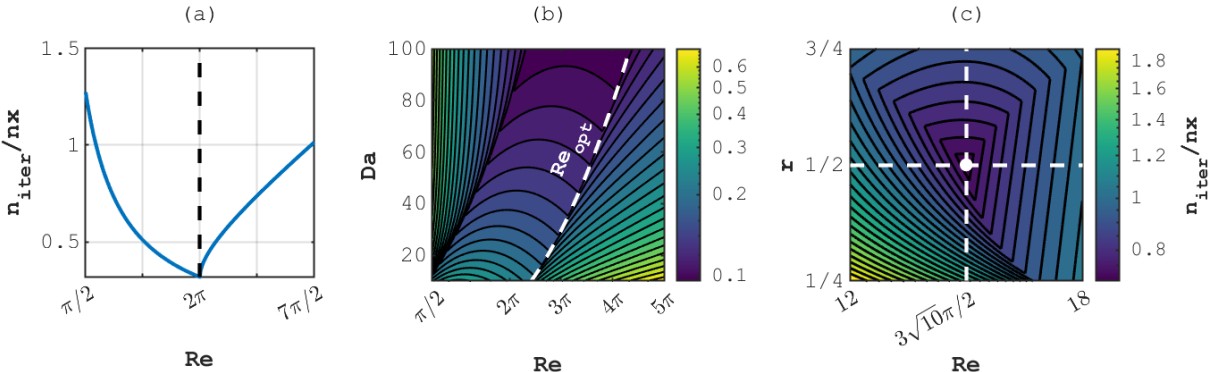

**Figure 1.** Number of iterations per grid point required for e-fold residual reduction. Panels **(a)**, **(b)** and **(c)** correspond to stationary diffusion, stationary reaction-diffusion, and incompressible 2D Stokes problems, respectively.

by the value corresponding to each grid point. If the distribution of $D$ is smooth, this approximation works well in practice and the number of iterations is close to the theoretically predicted value. However, particular care is needed when the distribution

of $D$ is discontinuous in space to avoid significantly reduced values of $\widetilde{C}$ to be required for convergence, ultimately leading to a much higher number of pseudo-transient iterations. We found that taking a local maximum of $D$ between neighbouring grid cells in the definitions of iteration parameters, Eq. (9) and (10), is sufficient to ensure optimal convergence. The per grid point local maximum selection thus effectively acts as a preconditioning technique.

For nonlinear and/or complex flow problems the corresponding optimal values of iteration parameters such as numerical

Reynolds number $\mathrm{Re}$ may be determined by systematic numerical experiments. In practice, optimal values for iteration parameters do not differ significantly from theoretical predictions derived for the linear case (see Sect. 2.1.3). Most importantly, the linear scaling of the method is still preserved. Also, the accelerated pseudo-transient method admits explicit numerical integration, and can be implemented with minimal modifications of the most simple pseudo-transient method.

### 2.1.3 Diffusion-reaction

The next example addresses stationary diffusion processes coupled with reaction. Here, we assume that reaction is described by the first-order kinetics law:

$$\tilde{\rho}\frac{\partial H}{\partial \tau} = -\nabla_k q_k - \rho\frac{H - H_{\mathrm{eq}}}{\theta_{\mathrm{k}}} \tag{12}$$

where $H_{\mathrm{eq}}$ is the value of $H$ at equilibrium, and $\theta_{\mathrm{k}}$ is a characteristic time of reaction. The flux $q$ in Eq. (12) is governed by either Eq. (2) or (6). The addition of a source term doesn't change the type of PDE involved in the method formulation, and the

iteration strategy based on the discretisation of Eq. (2) still exhibit a quadratic scaling. We therefore focus only on the analysis of the accelerated pseudo-transient method.





Equations (12) and (6) reduce to the following equation governing the evolution of $H$:

$$\tilde{\rho}\theta_{\mathrm{r}}\frac{\partial^2 H}{\partial \tau^2} + \left(\rho\frac{\theta_{\mathrm{r}}}{\theta_{\mathrm{k}}} + \tilde{\rho}\right)\frac{\partial H}{\partial \tau} = \nabla_k(D\nabla_k H) - \rho\frac{H - H_{\mathrm{eq}}}{\theta_{\mathrm{k}}} \tag{13}$$

The Eq. (13) differs from the damped wave equation (Eq. (7)) in that it includes the source term, and additional physical
parameters $\theta_{\mathrm{k}}$ and $\rho$. In the non-dimensional form, all parameters of Eq. (13) can be reduced to two non-dimensional numbers:
Re, defined equivalently to the stationary diffusion case, and $\mathrm{Da} = \rho L^2/D/\theta_{\mathrm{k}}$, a new parameter, which can be interpreted as
a Damköhler number characterising the ratio of characteristic diffusion time to reaction time scale. Contrary to the numerical
parameter Re, Da depends only on the physical parameters and cannot be arbitrarily specified.

We present the detailed dispersion analysis for the stationary diffusion-reaction problem in Sect. A3 of the Appendix. Param-
eters $\tilde{\rho}$ and $\theta_{\mathrm{r}}$ are defined according to Eq. (9) and (10), respectively, and by analogy to the stationary diffusion case. Optimal
values of Re depend now on the parameter Da:

$$\mathrm{Re}_{\mathrm{opt}} = \pi + \sqrt{\pi^2 + \mathrm{Da}} = \pi + \sqrt{\pi^2 + \frac{\rho L^2}{D\theta_{\mathrm{k}}}} \ . \tag{14}$$

We report the result of the dispersion analysis for the diffusion-reaction case as the number of iterations required for an e-fold
residual reduction $n_{\mathrm{iter}}$ per grid point $n_x$ as a function of Re and Da, highlighting $\mathrm{Re}_{\mathrm{opt}}$ as a function of Da by a dashed line
(Fig. 1b). In the limit $\mathrm{Da} \to 0$, i.e., when the characteristic time of reaction is infinitely large compared to the characteristic time
of diffusion, $\mathrm{Re}_{\mathrm{opt}} \to 2\pi$, which is the optimal value for the stationary diffusion problem discussed in Sect. 2.1.2. In that limit,
the number of iterations required for an e-fold residual reduction $n_{\mathrm{iter}}$ is also equivalent to the stationary diffusion problem.
However, as Da increases, the process becomes progressively more reaction-dominated and the pseudo-transient iterations
converge accordingly faster.

### 2.1.4 Transient diffusion

It is possible to apply the pseudo-transient method not only to the solution of stationary problems, but also to problems
including physical transient terms. This method is known in the literature as the "dual-time", or "dual-timestepping" method
(Gaitonde, 1998; Mandal et al., 2011).

According to the dual-time method, both physical and pseudo-time derivatives are present in the equation:

$$\tilde{\rho}\frac{\partial H}{\partial \tau} = -\nabla_k q_k - \rho\frac{\partial H}{\partial t} \tag{15}$$

The discretisation of the physical time derivative $\partial H/\partial t$ in Eq. (15) using a first-order backward Euler scheme, leads to:

$$\tilde{\rho}\frac{\partial H}{\partial \tau} = -\frac{\partial q}{\partial x} - \rho\frac{H - \widehat{H}}{\Delta t} \tag{16}$$

where $\widehat{H}$ is the distribution of $H$ at the explicit layer of the integration scheme, and $\Delta t$ is the physical time step. Comparing
Eq. (16) and (12) shows the two equations to be mathematically identical. Therefore, the optimal iteration parameters given by





Eq. (A15) apply for the transient diffusion as well. The $\mathrm{Da}$ parameter thus equals $\rho L^2 / D / \Delta t$ and characterises the fraction of the domain traversed by a particle transported by a diffusive flux during time $\Delta t$. The optimal value of $\mathrm{Re}$ is then defined as

$$\mathrm{Re}_{\mathrm{opt}} = \pi + \sqrt{\pi^2 + \frac{\rho L^2}{D \Delta t}} \; . \tag{17}$$

Frequently, modelling of certain processes requires relatively small time steps in order to capture important physical features, e.g., shear-heating induced strain localisation (Duretz et al., 2019a) or spontaneous flow localisation in porous media (Räss et al., 2019a). In that case, values of $\mathrm{Da}$ can be very large. Also, every step of numerical simulation serves as a good initial approximation to the next simulation step, thereby reducing error amplitude $E_1$ in Eq. (A16).

## 2.2 Incompressible viscous shear-driven Couette flow

Before considering incompressible Stokes equations, we present an illustrative example of shear-driven flow to demonstrate a similarity between already discussed cases addressing generalised diffusion and viscous fluid flow.

We here consider stationary fluid flow between two parallel plates separated by a distance $L$. We assume the absence of pressure gradients in directions parallel to the plates. In that case, Stokes equations reduce to the following system:

$$0 = \nabla_k \tau_{xk} \tag{18}$$

$$\frac{\tau_{xi}}{\mu_s} = \nabla_i v_x, \quad i, k \in \{y, z\} \tag{19}$$

where $\tau_{xi}$ is the deviatoric shear stress, $v_x$ is the velocity parallel to the plates, and $\mu_s$ the shear viscosity.

The steady state process described by Eq. (18) and (19) can be converted to a pseudo-transient process similar to the one presented in Sect. 2.1.2, by considering inertial term in the momentum equation (Eq. (18)) and Maxwell visco-elastic rheology as a constitutive relation for the viscous fluid (19):

$$\tilde{\rho} \frac{\partial v_x}{\partial \tau} = \nabla_k \tau_{xk} \tag{20}$$

$$\frac{1}{\widetilde{G}} \frac{\partial \tau_{xi}}{\partial \tau} + \frac{\tau_{xi}}{\mu_s} = \nabla_i v_x \; . \tag{21}$$

Here, $\tilde{\rho}$ and $\widetilde{G}$ are numerical parameters, interpreted as density and elastic shear modulus, respectively. The system of equations (Eq. (20) and (21)) is mathematically equivalent to the system of equations (Eq. (1) and Eq. (6)), describing pseudo-transient diffusion of the velocity field $v_x$. The relaxation time $\theta_{\mathrm{r}}$ represents in that case the Maxwell relaxation time, and is equal to $\mu_s / \widetilde{G}$.





### 2.3 Incompressible viscous Stokes equation

The next example addresses the incompressible creeping flow of a viscous fluid, described by Stokes equations:

$$0 = \nabla_j \left( \tau_{ij} - p\delta_{ij} \right) + f_i \tag{22}$$

$$0 = \nabla_k v_k \tag{23}$$

$$\frac{\tau_{ij}}{2\mu_s} = \frac{1}{2} \left( \nabla_i v_j + \nabla_j v_i \right) = \dot{\varepsilon}_{ij} \,, \tag{24}$$

where $\tau_{ij}$ is the deviatoric stress, $p$ is the pressure, $\delta_{ij}$ is the Kronecker delta, $f_i$ is the body forces, $v$ the velocity, and $\dot{\varepsilon}_{ij}$ is
the deviatoric strain rate.

Similarly to the shear-driven flow described in Sect. 2.2, a solution to the system (Eq. (22)–(24)) can be achieved by pseudo-transient time integration described by:

$$\tilde{\rho}\frac{\partial v_i}{\partial \tau} = \nabla_j \left( \tau_{ij} - p\delta_{ij} \right) + f_i \tag{25}$$

$$\frac{1}{\widetilde{K}}\frac{\partial p}{\partial \tau} = -\nabla_k v_k \tag{26}$$

$$\frac{1}{2\widetilde{G}}\frac{\partial \tau_{ij}}{\partial \tau} + \frac{\tau_{ij}}{2\mu_s} = \frac{1}{2} \left( \nabla_i v_j + \nabla_j v_i \right) \,. \tag{27}$$

Equation (25) and (26) now both include pseudo-time derivatives of velocity and pressure, and become an inertial and acoustic approximation to the momentum and mass balance equations, respectively. The additional parameter $\widetilde{K}$ arising in Eq. (26) can be interpreted as a numerical or pseudo- bulk modulus.

We use the primary, or P-wave velocity, as a characteristic velocity scale for the Stokes problem:

$$V_p = \sqrt{\frac{\widetilde{K} + 2\widetilde{G}}{\tilde{\rho}}} \,. $$

In addition to the non-dimensional numerical Reynolds number, here defined as $\mathrm{Re} = \tilde{\rho} V_p L/\mu_s$, we introduce the ratio between the bulk and shear elastic modulus $r = \widetilde{K}/\widetilde{G}$.

By analogy to previous cases, substituting $V_p = \widetilde{V}$ and solving for the numerical parameters $\tilde{\rho}$, $\widetilde{G}$ and $\widetilde{K}$ yields:

$$\tilde{\rho} = \mathrm{Re}\frac{\mu_s}{\widetilde{V} L} \,, \tag{28}$$

$$\widetilde{G} = \frac{\tilde{\rho} \widetilde{V}^2}{r + 2} \,, \tag{29}$$

$$\widetilde{K} = r\,\widetilde{G} \,. \tag{30}$$

Similarly to the diffusion-reaction problem studied in Sect. 2.1.3, there are two numerical parameters controlling the process. However, in Stokes equations, both parameters are purely numerical and could be tuned to achieve optimal convergence rate.





The dispersion analysis for 1D linear Stokes equations is detailed in Sect. A4 of the Appendix. We provide the resulting optimal values to converge the 2D Stokes problem:

$$\mathrm{Re_{opt}} = \frac{3\sqrt{10}}{2}\pi \, , \tag{31}$$

$$r_{\mathrm{opt}} = \frac{1}{2} \, , \tag{32}$$

because they differ from the 1D case values, and because we consider 2D and 3D Stokes formulation in the remaining of this study.

In the numerical experiments, we consistently observe faster convergence with slightly higher values of $r \approx 1$, likely caused by the fact that some of the assumptions made for 1D dispersion analysis do not transfer to the 2D formulation (Fig. 1c). Thus, the values presented in Eq. (31–32) should be only regarded as an optimal iteration parameters estimate.

### 2.4 Incompressible visco-elastic Stokes equation

The last example addresses the incompressible Stokes equations accounting for a physical visco-elastic Maxwell rheology:

$$\frac{1}{2G}\frac{\partial \tau_{ij}}{\partial t} + \frac{\tau_{ij}}{2\mu_s} = \frac{1}{2}\left(\nabla_i v_j + \nabla_j v_i\right) \, , \tag{33}$$

where $G$ is the physical shear modulus.

As in the transient diffusion case presented in the Sect. 2.1.4, the problem can be augmented by pseudo-transient time integration using a dual-timestepping approach:

$$\frac{1}{2\widetilde{G}}\frac{\partial \tau_{ij}}{\partial \tau} + \frac{1}{2G}\frac{\tau_{ij} - \hat{\tau}_{ij}}{\Delta t} + \frac{\tau_{ij}}{2\mu_s} = \frac{1}{2}\left(\nabla_i v_j + \nabla_j v_i\right) \, . \tag{34}$$

Collecting terms in front of $\tau_{ij}$ and ignoring $\hat{\tau}_{ij}$ because it doesn't change between successive pseudo-transient iterations, one can reduce the visco-elastic Stokes problem to the previously discussed viscous Stokes problem by replacing the viscosity in the Eq. (28) with the effective "visco-elastic" viscosity:

$$\mu^{\mathrm{ve}} = \left(\frac{1}{G\Delta t} + \frac{1}{\mu_s}\right)^{-1} \, . \tag{35}$$

The conclusions and optimal parameters' values presented in Sect. 2.3 remain thus valid for the visco-elastic rheology as well.

## 3 Performance and scaling

Assessing the performance of iterative stencil-based applications is two-fold, here reported in terms of algorithmic and implementation efficiency.

The accelerated pseudo-transient method provides an iterative approach that ensures linear scaling of the iteration count with increase in numerical grid resolution $n_x$ (Sect. 2) – the algorithmic scalability or performance. The major advantage in the design of such an iterative approach is it's concise implementation, extremely similar to explicit time integration schemes.





Explicit stencil-based applications, such as, e.g., elastic wave propagation, can show optimal performance and scaling on multi-GPU configurations because they can keep memory access to the strict minimum, leverage data locality and only require point-to-point communication (Podladtchikov and Podladchikov, 2013). We here follow a similar strategy.

We here introduce two metrics, the effective memory throughput $T_{\text{eff}}$ (early formulations of effective memory throughput
analysis are found, e.g., in Omlin et al. (2015a), Omlin et al. (2015b), and Omlin (2017)) and the parallel efficiency $E$ (Kumar et al., 1994; Gustafson, 1988; Eager et al., 1989). The effective memory throughput permits to assess the single-processor (GPU or CPU) performance and lets deduce potential room for improvement. The parallel efficiency permits to assess distributed memory scalability which may be hindered by inter-process communication, congestion of shared filesystems and other practical considerations from scaling on large supercomputers. We perform single-GPU problem size scaling bench-
marks to assess the optimal local problem size based on the $T_{\text{eff}}$ metric. We further use the optimal local problem size in weak scaling benchmarks to assess the parallel efficiency $E(N, P)$.

### 3.1 The effective memory throughput

Many-core processors such as GPUs are throughput-oriented systems that use their massive parallelism to hide latency. On the scientific application side, most algorithms require fewer floating point operations per second, FLOPS, compared to the amount
of numbers or bytes accessed from main memory, and thus are significantly memory bound. The FLOPS metric being no longer the most adequate for reporting the application performance (e.g. Fuhrer et al., 2018) in a majority of cases motivated us to develop a memory throughput-based performance evaluation metric, $T_{\text{eff}}$, to evaluate the performance of iterative stencil-based PDE solvers.

The effective memory access, $A_{\text{eff}}$ [GB], is the the sum of twice the memory footprint of the unknown fields, $D_{\text{u}}$, (fields that
depend on their own history and that need to be read from and written to every iteration) and the known fields, $D_{\text{k}}$, that do not change every iteration. The effective memory access divided by the execution time per iteration, $t_{\text{it}}$ [sec], defines the effective memory throughput, $T_{\text{eff}}$ [GB/s]:

$$A_{\text{eff}} = 2\,D_{\text{u}} + D_{\text{k}} \tag{36}$$

$$T_{\text{eff}} = \frac{A_{\text{eff}}}{t_{\text{it}}} \tag{37}$$

The upper bound of $T_{\text{eff}}$ is $T_{\text{peak}}$ as measured e.g. by McCalpin et al. (1995) for CPUs or a GPU analogue. Defining the $T_{\text{eff}}$ metric, we assume that i) we evaluate an iterative stencil-based solver, ii) the problem size is much larger than the cache sizes and iii) the usage of time blocking is not feasible or advantageous (which is a reasonable assumption for real-world applications). An important concept is not to include fields within the effective memory access that do not depend on their own history (e.g. fluxes); such fields can be re-computed on the fly or stored on-chip. Defining a theoretical upper bound for $T_{\text{eff}}$
that is closer to the real upper bound is work in progress (Omlin et al., 2021b).



## 3.2 The parallel efficiency

We employ the parallel efficiency metric to asses the scalability of the iterative solvers when targeting distributed memory configurations, such as multi-GPU settings. In a weak scaling configuration, i.e. where the global problem size and computing resources increase proportionally, the parallel efficiency $E(N, P)$ defines the ratio between the execution time of a single

process, $T(N, 1)$, and the execution time of $P$ processes performing the same number of iterations on an $P$-fold larger problem, $T(N \cdot P, P)$, where $N$ is the local problem size and $P$ is the number of parallel processes:

$$E(N, P) = \frac{T(N, 1)}{T(N \cdot P, P)} \ .$$  (38)

Distributed parallelisation permits to overcome limitations imposed by the available main memory of a GPU or CPU. It is particularly relevant for GPUs, which have significantly less main memory available than CPUs. Distributing work amongst

multiple GPUs, using e.g. the message passing interface (MPI), permits to overcome these limitations and requires parallel computing and supercomputing techniques. Parallel efficiency is a key metric in light of assessing the overall application performance as it ultimately ensures scalability of the pseudo-transient method.

## 4 The numerical experiments

We design a suite of numerical experiments to verify the scalability of the accelerated pseudo-transient method, targeting dif-

fusive processes and mechanics. We consider three distinct diffusion problems in one, two and three dimensions, that exhibit a diffusion coefficient being i) linear, ii) a step function with four orders of magnitude contrast and iii) a cubic power-law relation. We then consider mechanical processes using a velocity-pressure formulation to explore various limits including variable-viscosity incompressible viscous flow limit, accounting for a Maxwell visco-elastic shear rheology. To demonstrate the versatility of the approach, we tackle the nonlinear mechanical problem of strain localisation in two and three dimensions

considering an elasto-viscoplastic rheology (Sect. 8). Finally, we verify the robustness of the accelerated PT method by considering two parametric studies featuring different viscous Stokes flow patterns, and demonstrate the convergence of the method for viscosity contrasts up to 9 orders of magnitude.

### 4.1 Diffusive processes

We first consider time-dependent (transient) diffusion processes defined by Eq. (1) and (2), with the proportionality coefficient

$\rho = 1$. Practical applications often exhibit at least one diffusive component which can be either linear or nonlinear. Here, we consider linear and nonlinear cases representative of challenging configurations common to a broad variety of forward numerical diffusion-type of models:

1. The first case exhibits a linear constant (scalar) diffusion coefficient:

   $$D = 1 \ .$$  (39)

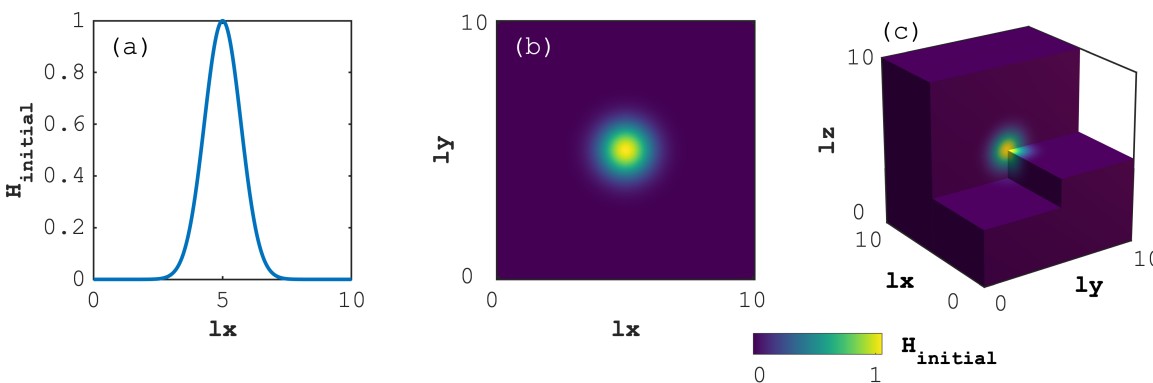

**Figure 2.** Initial distribution of $H$ for the **(a)** 1D, **(b)** 2D and **(c)** 3D time-dependent diffusion configurations.

2. The second case exhibits a spatially variable diffusion coefficient with a contrast of 4 orders of magnitude:

$$D = \begin{cases} 1 & \text{if } L < L_D , \\ 10^{-4} & \text{if } L >= L_D , \end{cases} \tag{40}$$

where $L$ is the domain extend in a specific dimension and $L_D$ the coordinate at which the transition occurs. Large contrasts in material parameters (e.g. permeability or heat conductivity) are common challenges solvers needs to handle when targeting real-world applications.

3. The third case exhibits a nonlinear power-law diffusion coefficient:

$$D = H^n , \tag{41}$$

where $n = 3$, a characteristic value in, e.g., soil and poro-mechanical applications to account for the porosity-permeability Carman-Kozeny (Costa, 2006) relation leading to the formation of solitary waves of porosity. Shallow ice approximation or nonlinear viscosity in power-law creep Stokes flow are other applications that exhibit effective diffusion coefficients
to be defined as power-law relations.

Practically, we implement the transient diffusion using the accelerated pseudo-transient method, solving Eq. (6) and (15) using a dual-time method (Sect. 2.1.4).

### 4.2   Mechanics

We secondly consider steady-state mechanical problems, defined by Eq. (22) and (23). In practice, we employ a velocity-
pressure formulation, which allows to also handle the incompressible flow limit. The rheological model builds upon an additive decomposition of the deviatoric strain rate tensor (Maxwell's model), given by Eq. (33).





In the following Sect. 5.2, the mechanical problem is solved in the incompressible limit and assuming a linear visco-elastic deviatoric rheology.

In the subsequent application (Sect. 8), the mechanical problem is solved in the compressible elasto-viscoplastic limit. Hence, the deviatoric rheological model neglects viscous flow and includes viscoplastic flow

$$\dot{\varepsilon}_{ij} = \dot{\varepsilon}_{ij}^{\mathrm{e}} + \dot{\varepsilon}_{ij}^{\mathrm{vp}} = \frac{1}{2G}\frac{\partial \tau_{ij}}{\partial t} + \dot{\lambda}\frac{\partial Q}{\partial \tau_{ij}},$$ (42)

where $\dot{\lambda}$ and $Q$ stand for the rate of the plastic multiplier and the plastic flow potential, respectively. A similar decomposition is assumed for the divergence of velocity in Eq. (23), which is no longer equal to zero in order to account for elastic and plastic bulk deformation:

$$\nabla_k v_k = \nabla_k v_k^{\mathrm{e}} + \nabla_k v_k^{\mathrm{vp}} = -\frac{1}{K}\frac{\partial p}{\partial t} - \dot{\lambda}\frac{\partial Q}{\partial p},$$ (43)

where $K$ stands for the physical bulk modulus.

In the inclusion parametric study described in Sect. 5.2, we consider the incompressible viscous Stokes flow limit, i.e. $K \to \infty$ and $G \to \infty$.

## 5 The model configurations

### 5.1 The diffusion model

We perform the three different diffusion experiments (see Sect. 4.1) on 1D, 2D and 3D computational domains (Fig. 2a, b and c, respectively). The only difference between the numerical experiments relies in the definition of the diffusion coefficient $D$. The non-dimensional computational domains are $\Omega_{1D} = [0, L_x]$, $\Omega_{2D} = [0, L_x] \times [0, L_y]$ and $\Omega_{3D} = [0, L_x] \times [0, L_y] \times [0, L_z]$, for 1D, 2D and 3D domains, respectively. The domain extend is $L_x = L_y = L_z = 10$. The initial condition $H_0$ consists of a Gaussian distribution of amplitude and standard deviation equal to one located in the domain's centre; in the 1D case:

$$H_0 = \exp\left(-\left(x_{\mathrm{c}} - 0.5\,L_x\right)^2\right),$$ (44)

where $x_{\mathrm{c}}$ is the vector containing the discrete 1D coordinates of the cell centres. The 2D and 3D cases are done by analogy and contain the respective terms for the $y$ and $z$ directions. We impose Dirichlet boundary conditions such that $H = 0$ on all boundaries. We simulate a total non-dimensional physical time of 1 performing 5 implicit time steps of $\Delta t = 0.2$.

### 5.2 The Stokes flow model

We perform the visco-elastic Stokes flow experiments (see Sect. 4.2) on 2D and 3D computational domains (Fig. 3a and b, respectively). The non-dimensional computational domains are $\Omega_{2D} = [0, L_x] \times [0, L_y]$ and $\Omega_{3D} = [0, L_x] \times [0, L_y] \times [0, L_z]$ for 2D and 3D domains, respectively. The domain extend is $L_x = L_y = L_z = 10$. We define as initial condition a circular (2D) or spherical (3D) inclusion of radius $r = 1$ centred at $L_x/2,\ L_y/2$ (2D) and $L_x/2,\ L_y/2,\ L_z/2$ (3D) featuring 3 order of



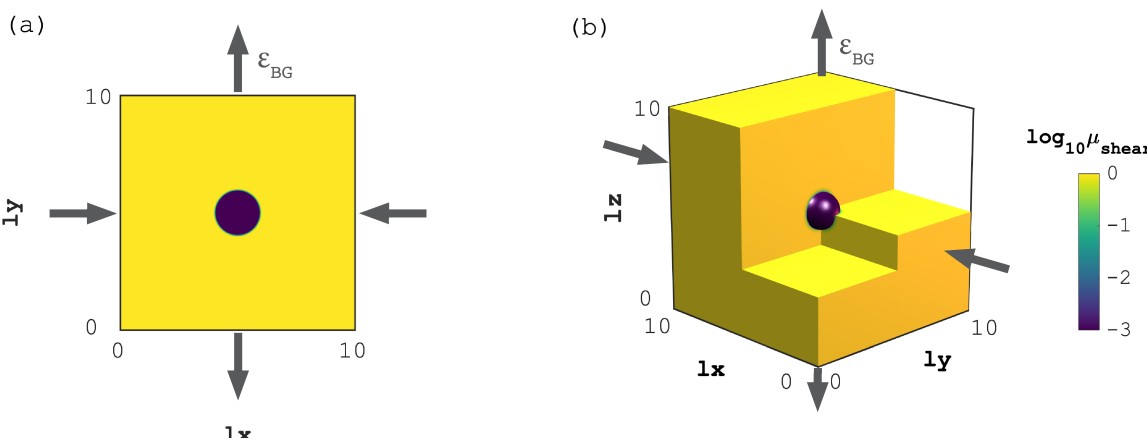

**Figure 3.** Initial shear viscosity $\mu_{\mathrm{shear}}$ distribution for the **(a)** 2D and **(b)** 3D visco-elastic Stokes flow configuration, respectively.

magnitude lower shear viscosity $\mu_{\mathrm{s}}^{\mathrm{inc}} = 10^{-3}$ compared to the background value $\mu_{\mathrm{s}}^0 = 1$ (Fig. 3). We then perform 10 explicit diffusion steps of the viscosity field $\mu_{\mathrm{s}}$ to account for smoothing introduced by commonly employed advection schemes (e.g. markers-in-cells, semi-Lagrangian or weighted ENO). We define a uniform and constant elastic shear modulus $G = 1$ and chose the physical time step $\Delta t = \mu_{\mathrm{s}}^0/G/\xi$ to satisfy a visco-elastic Maxwell relaxation time of $\xi = 1$. We impose pure shear boundary conditions; we apply compression in the $x$-direction and extension in the vertical ($y$ in 2D, $z$ in 3D) direction with a

background strain rate $\varepsilon_{\mathrm{BG}} = 1$. For the 3D case, we apply no inflow/outflowin the $y$-direction (Fig. 3b). All models boundaries are free to slip. We perform a total of 5 implicit time steps to resolve visco-elastic stress build-up.

We further perform a series of viscous Stokes numerical experiments in 2D (see Sect. 4.2) to analyse the dependence of the optimal iteration parameters on the material viscosity contrast and the volume fraction of the material with lower viscosity. The non-dimensional computational domain is $\Omega_{2D} = [0, L_x] \times [0, L_y]$. We define as initial condition a number $n_{\mathrm{inc}}$ of circular

inclusions semi-uniformly distributed in the domain. The viscosity in the inclusions is $\mu_{\mathrm{s}}^{\mathrm{inc}}$ and the background viscosity is $\mu_{\mathrm{s}}^0 = 1$.

In the parametric study, we vary the number of inclusions $n_{\mathrm{inc}}$, the inclusion viscosity $\mu_{\mathrm{s}}^{\mathrm{inc}}$, and the iteration parameter $\mathrm{Re}$. We consider a uniform 3D grid of parameter values, numerically calculating for each combination of these the steady-state distribution of stresses and velocities.

We consider two different problem setups that correspond to important edge cases. The first setup addresses the shear-driven flow where the strain rates are assumed to be applied externally via boundary conditions. This benchmark might serve as a basis for the effective media properties calculation. The second setup addresses the gravity-driven flow with buoyant inclusions. This benchmark is relevant for geophysical applications, e.g. modelling magmatic diapirism or melt segregation, where the volumetric effect of melting leads to the development of either the Raileigh-Taylor instability or compaction instability,

respectively.





In the first setup, we specify pure-shear boundary conditions similarly to the singular inclusion case described in Sect. 5.2. The body forces $f_i$ are set to zero in Eq. (22).

In the second setup, we specify the free-slip boundary conditions, which corresponds to setting the background strain rate $\varepsilon_{\mathrm{BG}}$ to 0. We model buoyancy using the Boussinesq approximation: the density differences are accounted for only in the body forces. We set $f_x = 0$, $f_y = -\rho g$. We set $\rho g^0 = 1$, $\rho g^{\mathrm{inc}} = 0.5$.

## 6 Discretisation

We discretise the systems of partial differential equations (Sect. 4) using the finite-difference method on a regular Cartesian staggered grid. For the diffusion process, the quantity being diffused and the fluxes are located at cell centre and at cell interfaces, respectively. For the Stokes flow, pressure, normal stresses and material properties (e.g. viscosity) are located at cell centres while velocities are located at cell interfaces. Shear stress components are located at cell vertices. The staggering relies on second-order conservative finite-differences (Patankar, 1980; Virieux, 1986; McKee et al., 2008) ensuring also the Stokes flow to be inherently devoid of oscillatory pressure modes (Shin and Strikwerda, 1997).

The diffusion process and the visco-elastic Stokes flow include physical time evolution. We implement a backward Euler time integration within the pseudo-transient solving procedure (see Sect. 2) and do not assess higher order schemes as such considerations go beyond the scope of this study.

We converge in all simulations the scaled and normalised L2-norm of the residuals, $||R||_{\mathrm{L2}}/\sqrt{n_R}$, where $n_R$ stands for the number of entries of $R$, for each physical time step to a nonlinear absolute tolerance of $\mathrm{tol}_{\mathrm{nl}} = 10^{-8}$ within the iterative pseudo-transient procedure (absolute and relative tolerances being comparable given the non-dimensional form of the example we here consider).

The 46-lines code fragment (Fig. 4) informs about the concise implementation of the accelerated PT algorithm, here for the 1D nonlinear power-law diffusion case ($D = H^3$). Besides the initialisation part (lines 3-22), the core of the algorithm is contained in no more than 20 lines (lines 23-43). The algorithm is implemented as two nested (pseudo-)time loops, referred to as "dual-time"; the outer loop advancing in physical time, the inner loop converging the implicit solution in pseudo-time. The nonlinear term, here the diffusion coefficient $D$, is explicitly evaluated within the single-loop iterative procedure, removing the need of performing nonlinear iterations on top of a linear solve (e.g. Brandt, 1977; Trottenberg et al., 2001; Hackbusch, 1985). This single loop local linearisation shows, in practice, lower iteration count when compared against global linearisation (nested loops). Note that a relaxation of nonlinearities can be implemented in straight-forward fashion in case the nonlinear term hinders convergence (see implementation details in , e.g., Räss et al. (2020, 2019a); Duretz et al. (2019b)). The iteration parameters are evaluated locally which ensures scalability of the approach and removes the need of performing global reductions, costly in parallel implementation. Note that the numerical iteration parameters $\tilde{\rho}$ and $\theta_{\mathrm{r}}$, arising from the finite-difference discretisation of pseudo-time derivatives in Eq. (16) and (6),

$$\tilde{\rho}\frac{\partial H}{\partial \tau} \approx \tilde{\rho}\frac{H^k - H^{k-1}}{\Delta \tau}, \quad \theta_{\mathrm{r}}\frac{\partial q_i}{\partial \tau} \approx \theta_{\mathrm{r}}\frac{q_i^k - q_i^{k-1}}{\Delta \tau}, \tag{45}$$





```
# [...]
@views function diffusion_1D(; implicit=false, do_viz=true)
# Physics
lx      = 10.0      # domain size
ttot    = 1.0       # total simulation time
dt      = 0.2       # physical time step
# Numerics
nx      = 512       # numerical grid resolution
tol     = 1e-8      # tolerance
CFL     = 0.8       # CFL number: 1.0 implicit, 0.8 explicit
ε       = 1e-2      # small number for explicit stability
# [...]
# Derived numerics
dx      = lx / nx   # grid size
Vpdτ    = CFL * dx
xc      = LinRange(dx / 2, lx - dx / 2, nx)
# Array allocation
# [...]
# Initial condition
H0      = exp.(-(xc .- lx / 2).^2)
# [...]
# Physical time loop
while it < nt
iter = 0; err = 2 * tol
# Pseudo-transient iteration
while err > tol && iter < itMax
# Diffusion coefficient
D         .= H.^3
# Assign iter params
Re        .= π .+ sqrt.(π^2 .+ lx^2 ./ max.(D,ε) ./ dt)
θr_dτ     .= lx ./ Vpdτ ./ av(Re)
dτ_ρ      .= Vpdτ .* lx ./ max.(D[2:end-1],ε) ./ Re[2:end-1]
# PT updates
# [...]
qHx       .= qHx         .+  1.0 ./ θr_dτ .* ( .-qHx .-av(D) .* diff(H) ./ dx )
H[2:end-1] .= H[2:end-1] .+  dτ_ρ        .* ( .-(H[2:end-1] .- Hold[2:end-1]) ./ dt .- diff(qHx) ./ dx )
# Check errors
# [...]
end
# Update H
Hold .= H
ittot += iter; it += 1; t += dt
end
# [...]
return
end
```

**Figure 4.** Numerical Julia implementation of the 1D nonlinear diffusion case $D = H^3$. Lines marked with `# [...]` refer to skipped lines. See diff_1D_nonlin_simple.jl script located in the scripts folder in PseudoTransientDiffusion.jl on GitHub for full code.

where $k$ is the current pseudo-time iteration index, always occur in combination with $\Delta\tau$. Since we are not interested in the evolution of pseudo-time or the particular values of iteration parameters, it is possible to combine them in the implementation.

495 We therefore introduce the variables $d\tau\_\rho = \Delta\tau/\rho$ and $\theta r\_d\tau = \theta r/\Delta\tau$. Using the new variables helps to avoid specifying the value of $\Delta\tau$ which could otherwise be specified arbitrarily. The 2-lines of physics, namely the PT updates, are here evaluated in an explicit fashion. Alternatively, one could solve for `qHx` and `H` assuming their values in the residual –the terms contained in the right-most parenthesis– to be new instead of current resulting in an implicit update. Advantages rely in enhanced stability (`CFL` on line 10 could be set to 1) and removes the need of defining a small number ($\varepsilon$ in the iteration parameters

500 definition) to prevent division by 0. The implicit approach is implemented as alternative in the full code available online in the PseudoTransientDiffusion.jl GitHub repository.



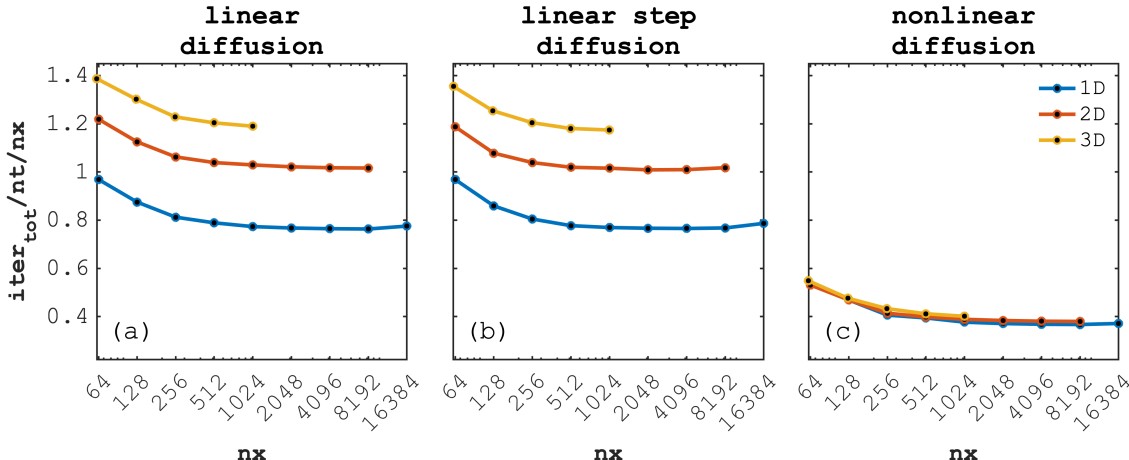

**Figure 5.** Iteration count scaled by the number of time steps $n_t$ and by the number of grid cells in the $x$-direction $n_x$ as function of $n_x$, comparing 1D, 2D and 3D models for the three different diffusion configurations, **(a)** linear diffusion, **(b)** linear step diffusion and **(c)** nonlinear diffusion, respectively.

## 6.1 The Julia multi-XPU implementation

We use the Julia language (Bezanson et al., 2017) to implement the suite of numerical experiments. Julia's high-level abstractions, multiple dispatch and meta-programming capabilities make it amenable to portability between backends (e.g. multi-core CPUs and Nvidia or AMD GPUs). Also, Julia solves the two-language problem making it possible to fuse prototype and production applications into a single one being both high-level and performance oriented – ultimately increasing productivity.

We use the ParallelStencil.jl (Omlin et al., 2021b) and ImplicitGlobalGrid.jl (Omlin et al., 2021a) Julia packages we developed as building blocks to implement the diffusion and Stokes numerical experiments. ParallelStencil.jl permits to write architecture-agnostic high-level code for parallel high-performance stencil computations on GPUs and CPUs – here referred to as XPUs. Performance similar to native CUDA C/C++ (Nvidia GPUs) or HIP (AMD GPUs) can be achieved. ParallelStencil.jl seamlessly composes with ImplicitGlobalGrid.jl, which allows for distributed parallelisation of stencil-based XPU applications on a regular staggered grid. In addition, ParallelStencil.jl enables hiding communication behind computation, where the communication package used can, a priori, be any package that lets the user control when communication is triggered. The communication and computation overlap approach splits local domain calculations in two regions, boundary regions and inner region, the latter containing majority of the local domain's grid cells. After successful completion of the boundary region computations, halo update (using e.g. point-to-point MPI) overlaps with inner point computations. Selecting the appropriate width of the boundary region permits to fine-tune optimal hiding of MPI communication (Räss et al., 2019c; Alkhimenkov et al., 2021b).





In the present study, we focus on using ParallelStencil.jl with the CUDA.jl backend to target Nvidia GPUs (Besard et al.,
2018, 2019), and ImplicitGlobalGrid.jl which relies on MPI.jl (Byrne et al., 2021), Julia's MPI wrappers, to enable distributed
memory parallelisation.

## 7  Results

We here report the performance of the accelerated pseudo-transient Julia implementation of the diffusion and the Stokes flow
solvers targeting Nvidia GPUs using ParallelStencil.jl's CUDA backend. For both physical processes, we analyse the iteration
count as function of number of grid cells (i.e. the algorithmic performance), the effective memory throughput $T_{\mathrm{eff}}$ [GB/s]
(performing a single device (GPU) problem size scaling), and the parallel efficiency $E$ (multi-GPU weak scaling).

We report the algorithmic performance as the iteration count per number of physical time steps normalised by the number
of grid cells in the $x$-direction. We do not normalise by the total number of grid cells in order to report the one-dimensional
scaling even for 2D or 3D implementation. We motivate our choice as it permits a more accurate comparison to analytically
derived results and leave it to the reader to appreciate the actual quadratic and cubic dependence of the normalised iteration
count if using the total number of grid cells in 2D and 3D configurations, respectively.

### 7.1  Solving the diffusion equation

We report a normalised iteration count per total number of physical time steps $n_{\mathrm{t}}$ per number of grid cells in the $x$-direction
$n_x$, $\mathrm{iter_{tot}/n_t/n_x}$, for the 1D, 2D and 3D implementations of the diffusion solver for the linear, step function and nonlinear
case (Fig. 5a, b and c, respectively) relating to the spatial distribution of $H$ after 5 implicit time steps (Fig. 6a-c, d-f and g-i,
respectively). All three different configurations exhibit a normalised number of iterations per time step per number of grid
cells close to 1 for the lowest resolution of $n_x = 64$ grid cells. The normalised iteration count drops with increase in numerical
resolution (increase in number of grid cells) suggesting a super-linear scaling. We observe similar behaviour when increasing
the number of spatial dimensions while solving the identical problem; 3D calculations are more efficient on a given number of
grid cells $n_x$ compared to the corresponding 1D or 2D calculations.

Interesting to note that the diffusion solver with nonlinear (power-law) diffusion coefficient reports the lowest normalised
iteration count for all three spatial dimension implementations, reaching the lowest number ($> 0.4$) of normalised iteration
count in the 3D configuration (Fig. 5c). The possible explanation of lower iteration counts for the nonlinear problem is that by
the nature of the solution, the distribution of the diffused quantity $H$ at $t = 1$ is much closer to the initial profile than in the
linear case. Therefore, at each time step, the values of $H$ are closer to the values of $H$ at the next time step and thus serve as
a better initial approximation. Both the diffusion with linear (Fig. 5a) and step function as diffusion coefficient (Fig. 5b) show
similar trend in their normalised iteration count with values decreasing while increasing the number of spatial dimensions.



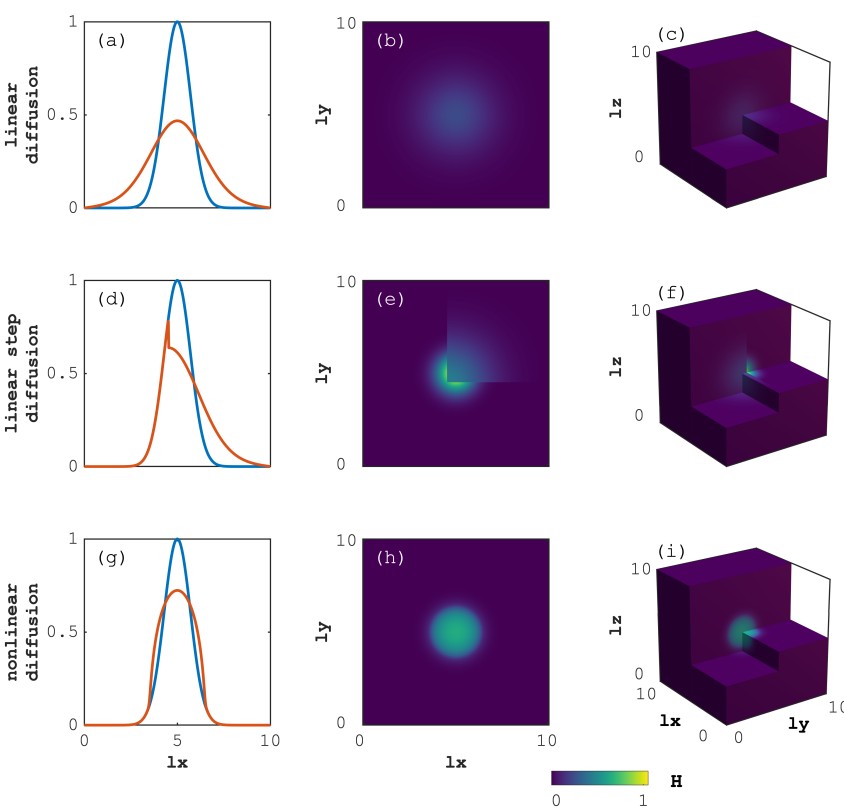

**Figure 6.** Model output for the **(a, d, g)** 1D, **(b, e, h)** 2D and **(c, f, i)** 3D time-dependent diffusion of a quantity $H$. The upper **(a-c)**, centre **(d-f)** and lower **(g-i)** panels refer to the diffusion with linear, linear step and nonlinear (power-law) diffusion coefficient, respectively. For the 1D case, the blue and red lines represent the initial and final distribution, respectively. The colormap (2D and 3D) relates to the $y$-axis (1D).

## 7.2 Solving visco-elastic Stokes flow

We further report the normalised iteration count per total number of physical time steps $n_t$ per number of grid cells in the $x$-direction $n_x$, $\mathrm{iter_{tot}}/\mathrm{n_t}/\mathrm{n_x}$, for the 2D and 3D implementations of the visco-elastic Stokes solver (Fig 7) relating to the spatial distribution of vertical velocity (deviation from background) $\Delta V_{\mathrm{vertical}}$, pressure $P$ and shear stress $\tau_{\mathrm{shear}}$ after 5 implicit time steps (Fig. 8a-b, c-d and e-f, respectively). Both 2D and 3D visco-elastic Stokes flow exhibit a normalised number of iterations per time step per number of grid cells close to 10 for the lowest resolution of $n_x = 63$ grid cells. The normalised iteration count drops with increase in numerical resolution (increase in number of gird cells) suggesting a super-linear scaling. We observe similar behaviour when increasing the number of spatial dimensions from 2D to 3D, while solving the identical problem; 3D calculations are more efficient on a given number of grid cells $n_x$ compared to the corresponding 2D calculations, which is in accordance with results for the various diffusion solver configurations.

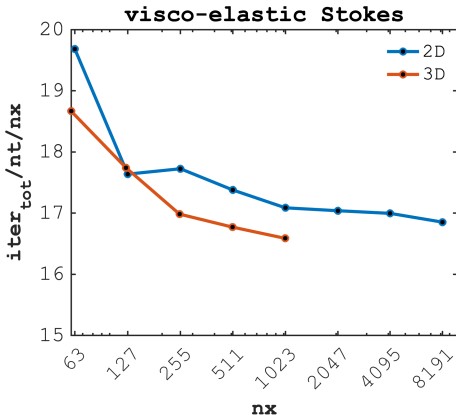

**Figure 7.** Iteration count scaled by the number of time steps $n_t$ and by the number of grid cells in the $x$-direction $n_x$ as function of $n_x$, comparing 2D and 3D visco-elastic Stokes flow containing an inclusion featuring a viscosity contrast ($\mu_0/\mu_i = 10^3$).

The visco-elastic Stokes flow scaling results confirm the trend reporting a decrease of the normalised iteration count with increasing the numerical resolution (number of grid cells). Interesting to note that the accelerated pseudo-transient implemen-
tation of the 3D visco-elastic Stokes flow featuring 3 orders of magnitude viscosity contrast ($\mu_s^0/\mu_s^{inc} = 10^3$) only requires less than 17 normalised iterations when targeting resolutions of $1023^3$ (Fig. 7).

### 7.3  Performance

We use the effective memory throughput $T_{\text{eff}}$ [GB/s] and the parallel efficiency $E(N, P)$ to asses the implementation perfor-
mance of the accelerated pseudo-transient solvers, as motivated in Sect. 3. We perform the single-GPU problem size scaling
and the multi-GPU weak scaling tests on different Nvidia GPU architectures, namely the "data-centre" GPUs, Tesla P100 (Pas-
cal - PCIe), Tesla V100 (Volta - SXM2) and Tesla A100 (Ampere - SXM2). We run the weak scaling multi-GPU benchmarks
on the *Piz Daint* supercomputer, featuring up to 5704 Nvidia Tesla P100 GPUs, at the Swiss National Supercomputing Centre
CSCS, on the *Volta* node on the *Octopus* supercomputer, featuring 8 Nvidia Tesla V100 GPUs with high-throughput (300
GB/s) SXM2 interconnect, at the Swiss Geocomputing Centre, University of Lausanne, and on the *Superzack* node, featuring 8
Nvidia Tesla A100 GPUs with high-throughput (300 GB/s) SXM2 interconnect, at the Lab. Hydraulics, Hydrology, Glaciology
(VAW), ETH Zurich.

We asses the performance of the 2D and 3D implementation of the nonlinear diffusion solver (power-law diffusion coeffi-
cient) and the visco-elastic Stokes flow solver, respectively. We perform single-GPU problem size scaling tests for both the 2D
and 3D solvers' implementation, and multi-GPU weak scaling tests for the 3D solvers' implementation only. We report the
mean performance out of 5 executions, if applicable.



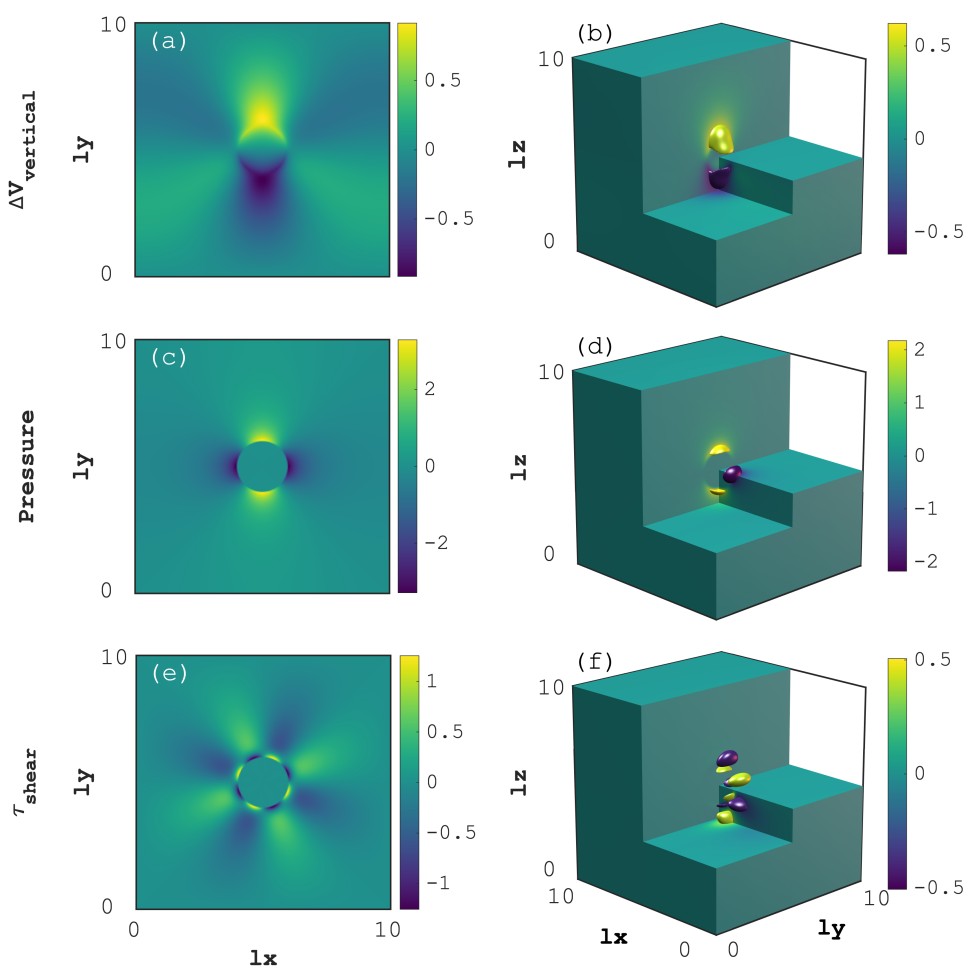

**Figure 8.** Model output for the **(a, c, e)** 2D and **(b, d, f)** 3D visco-elastic Stokes flow containing an inclusion featuring a viscosity contrast $(\mu_0/\mu_i = 10^3)$. The upper **(a-b)**, centre **(c-d)**, and lower **(e-f)** panels depict the deviation from background vertical velocity $\Delta V_{\mathrm{vertical}}$, the dynamic pressure $P$ and deviatoric shear stress $\tau_{\mathrm{shear}}(\tau_{xy}$ in 2D, $\tau_{xz}$ in 3D) distribution, respectively.

### 7.3.1 Single-GPU problem size scaling and effective memory throughput

The 2D and 3D nonlinear diffusion solver single-GPU problem size scaling benchmarks achieve similar effective memory throughput on the targeted GPU architectures relative to their respective peak values $T_{\mathrm{peak}}$; values of $T_{\mathrm{eff}}$ for the 2D implementation being slightly higher than the 3D ones for the Volta and Pascal architectures, but similar for Ampere one. This

discrepancy is expected and may be partly explained by an increase in cache-misses when accessing $z$-direction neighbours

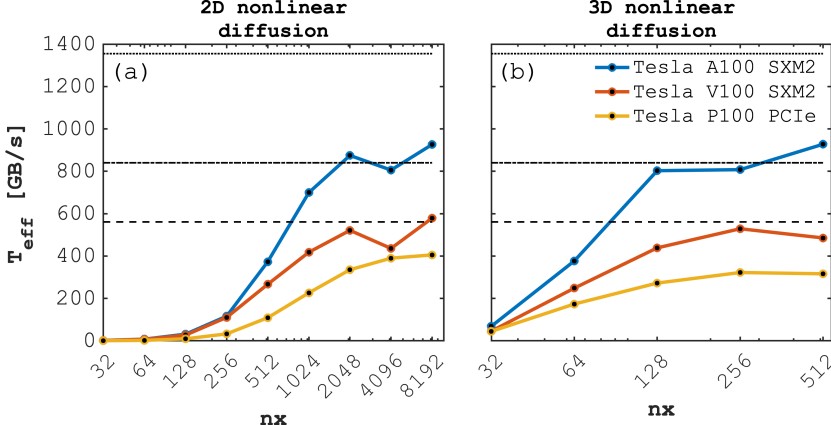

**Figure 9.** Effective memory throughput $T_{\text{eff}}$ [GB/s] for the **(a)** 2D and **(b)** 3D nonlinear diffusion Julia GPU implementations using Parallel-Stencil.jl executed on various Nvidia GPUs (Tesla A100 SXM2, Tesla V100 SXM2 and P100 PCIe). The dashed lines report the measured peak memory throughput $T_{\text{peak}}$ (1355 GB/s Tesla A100, 840 GB/s Tesla V100, 561 GB/s Tesla P100).

which are $n_x \cdot n_y$ grid cells away in main memory. We achieve in 2D $T_{\text{eff}} \approx 920$ GB/s, 590 GB/s and 400 GB/s on the Tesla A100, Tesla V100 and the Tesla P100, respectively (Fig. 9a). In 3D, we achieve $T_{\text{eff}} \approx 920$ GB/s, 520 GB/s and 315 GB/s on the Tesla A100, Tesla V100 and the Tesla P100, respectively (Fig. 9b).

For the analogous visco-elastic Stokes flow single-GPU problem size scaling tests, we report also higher $T_{\text{eff}}$ values for the
2D compared to the 3D implementation for all three targeted architectures. We achieve in 2D $T_{\text{eff}} \approx 930$ GB/s, 500 GB/s and 320 GB/s on the Tesla A100, Tesla V100 and the Tesla P100, respectively (Fig. 10a). In 3D, we achieve $T_{\text{eff}} \approx 730$ GB/s, 350 GB/s and 230 GB/s on the Tesla A100, Tesla V100 and the Tesla P100, respectively (Fig. 10b). Increased neighbouring access and overall more derivative evaluations may explain the slightly lower effective memory throughput of the visco-elastic Stokes flow solver when compared to the nonlinear diffusion solver.

For reference, the dashed lines (Figs. 9 and 10) represent the peak memory throughput $T_{\text{peak}}$ [GB/s] one can achieve on a specific GPU architecture, which is a theoretical upper bound of $T_{\text{eff}}$.

### 7.3.2 Weak scaling and parallel efficiency

We assess the parallel efficiency of the 3D nonlinear diffusion and visco-elastic Stokes flow solver multi-GPU implementation performing a weak-scaling benchmark. We use (per GPU) local problem size of $512^3$ for the nonlinear diffusion, and $383^3$ and
$511^3$ for the visco-elastic Stokes flow on the Pascal and Tesla architectures, respectively. Device RAM limitations prevent to solve a larger local problem in the latter case. The 3D nonlinear diffusion solver achieves a parallel efficiency $E$ of 97% on 8 Tesla A100 and V100 SXM2 and 98% on 2197 Tesla P100 GPUs (Fig. 11a). The visco-elastic Stokes flow solver achieves a parallel efficiency $E$ of 99% on 8 Tesla A100 SXM2, and 96% on 8 Tesla V100 SXM2 and on 2197 Tesla P100 GPUs (Fig. 11b), respectively. 2197 GPUs represent a 3D Cartesian topology of $13^3$, resulting in global problem sizes of $6632^3$ and $4995^3$



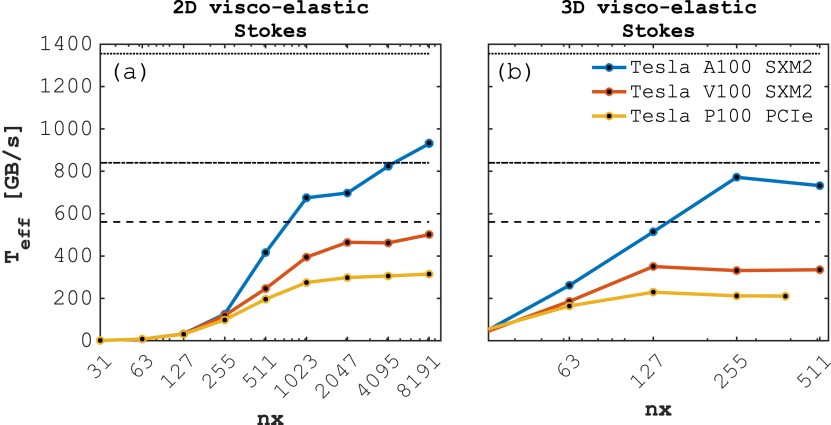

**Figure 10.** Effective memory throughput $T_{\text{eff}}$ [GB/s] for the **(a)** 2D and **(b)** 3D the visco-elastic Stokes flow Julia GPU implementations using ParallelStencil.jl executed on various Nvidia GPUs (Tesla A100 SXM2, Tesla V100 SXM2 and P100 PCIe). The dashed lines report the measured peak memory throughput $T_{\text{peak}}$ (1355 GB/s Tesla A100, 840 GB/s Tesla V100, 561 GB/s Tesla P100).

grid cells, for the nonlinear diffusion (291 giga DoFs) and visco-elastic Stokes flow (1.2 tera DoFs), respectively. In terms of cumulative effective memory throughput $T_{\text{eff}}$, the 3D diffusion and Stokes flow solver achieve 679 TB/s and 444 TB/s, respectively. This near petabyte per second effective throughput reflects the impressive memory bandwidth exposed by GPUs and requires efficient algorithms to leverage it.

We emphasise that we follow a *strict definition of parallel efficiency*, where the runtimes of the multi-XPU implementations
are to be compared against the best known single-XPU implementation. As a result, the reported parallel efficiency is also with a single GPU below 100%, correctly showing that the implementation used for distributed parallelisation performs slightly less good then the best known single-GPU implementation. This small performance loss emerges from the computation splitting in boundary and inner regions required by the hide communication feature. Parallel efficiency close to 100% is important to ensure weak scalability of numerical applications when executed on a growing number of distributed memory processes $P$, the
path to leverage current and future supercomputers' exascale capabilities.

### 7.4 Multiple inclusions parametric study

We perform a multiple inclusions benchmark to assess the robustness of the developed accelerated PT method. We vary the viscosity contrast from 1 to 9 orders of magnitude to demonstrate the successful convergence of iterations even for extreme cases that might arise in geophysical applications such as strain localisation. Further, we vary the number of inclusions from 1
to 46 to verify the independence of convergence on the "internal geometry" of the problem. For each combination of viscosity ratio and number of inclusions, we perform a series of simulations varying the iteration parameter Re to assess the influence of the problem configuration on its optimal value and verify if the analytical prediction obtained by the dispersion analysis remains valid over the considered parameter range.



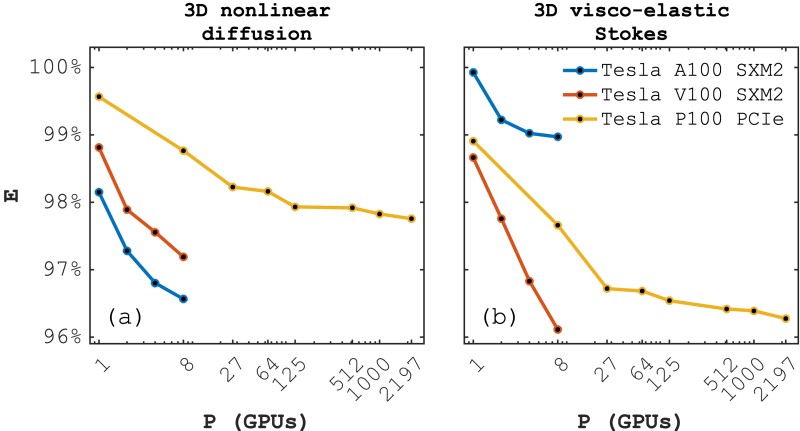

**Figure 11.** Parallel efficiency $E$ as function of 1-8 Tesla A100 SXM2, 1-8 Tesla V100 SXM2, and 1-2197 Tesla P100 Nvidia GPUs ($P$). Weak scaling benchmark for **(a)** the 3D nonlinear diffusion solver and **(b)** the 3D visco-elastic Stokes flow solver based on a 3D Julia implementation using ParallelStencil.jl and ImplicitGlobalGrid.jl.

For this parametric study, we considered a computational grid consisting of $n_x \times n_y = 2048^2$ cells. At lower resolutions the
convergence deteriorates, resulting in non-converging large viscosity contrasts configurations. The high grid resolution is thus
necessary for resolving the small-scale details of the flow. We also adjust the nonlinear tolerance for the iterations to $10^{-5}$ and
$10^{-3}$ for momentum and mass balance, respectively, given our interest in relative dependence of iteration counts on iteration
parameter $\mathrm{Re}$.

Figure 12 depicts the results for the shear-driven flow case. For a single inclusion (Fig. 12a), the optimal value of iteration
parameter $\mathrm{Re}$ does not differ significantly from the one reported by Eq. (31). Moreover, the theoretical prediction for $\mathrm{Re}$
remains valid for all viscosity contrasts considered in the study.

For problem configurations involving 14 and 46 inclusions (Fig. 12b and c, respectively), the minimal number of iterations
is achieved for values of $\mathrm{Re}$ close to the theoretical prediction only for viscosity contrast of 1 order of magnitude. For larger
viscosity contrast, the optimal value of $\mathrm{Re}$ appears to be lower then theoretically predicted, and the overall iteration count is
significantly higher. These iteration counts reach $40n_x$ at the minimum among all values of $\mathrm{Re}$ for a given viscosity ratio, and
$> 50n_x$ for non-optimal values of $\mathrm{Re}$.

For buoyancy-driven flow (Fig. 13), the convergence of iterations is less sensitive to both the number of inclusions and the
viscosity ratio. The observed drift in the optimal value of $\mathrm{Re}$ could be partly attributed to the lack of a good preconditioning
technique. In this study, we specify the local iteration parameters in each grid cell based on the values of material parameters,
which could be regarded as a form of diagonal (Jacobi) preconditioning. This choice is motivated by the parallel scalability
requirements of GPU and parallel computing. Even without employing more advanced preconditioners, our method remains
stable and successfully converges for viscosity contrasts up to 9 orders of magnitude, though at the cost of increased number





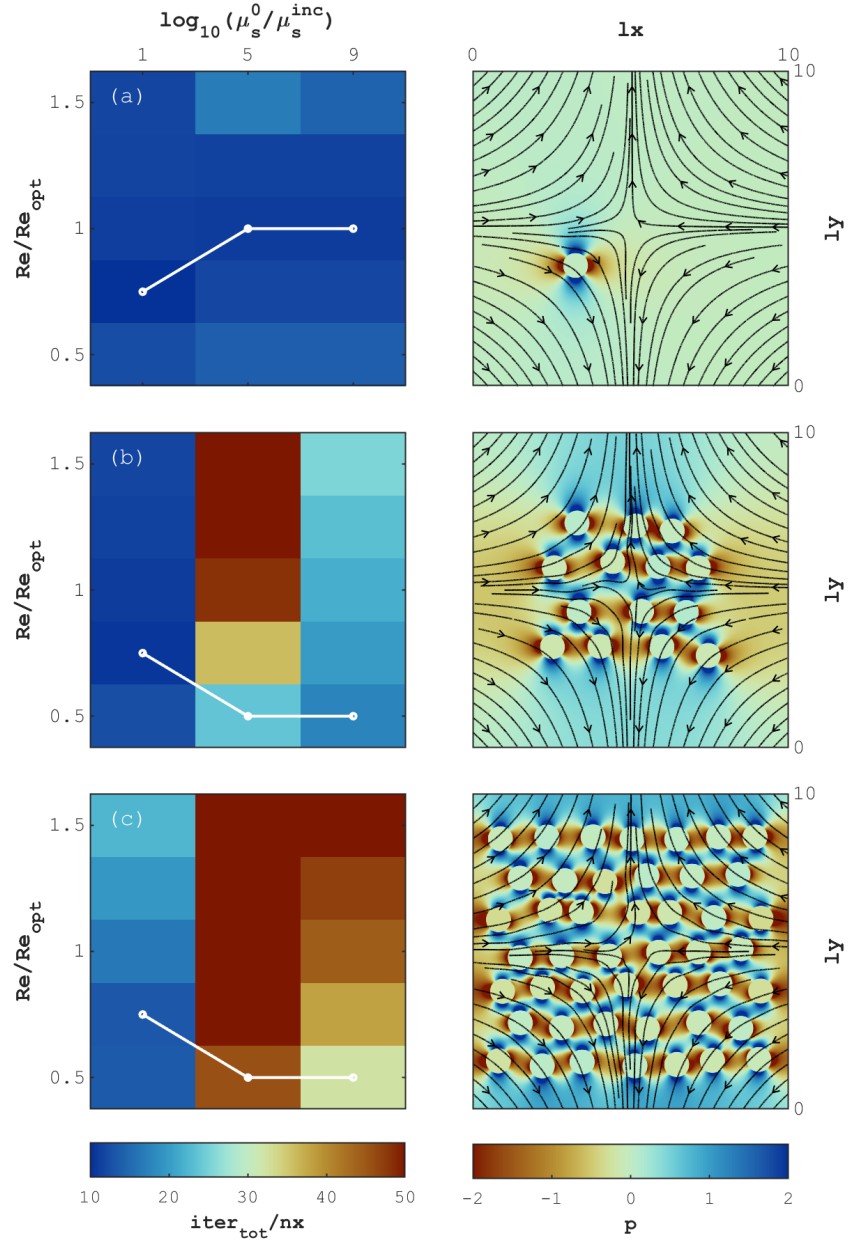

**Figure 12.** Pure shear-driven flow. The left column reports the number of iterations for different values of viscosity ratio $\mu_s^0/\mu_s^{inc}$ and ratio of numerical Reynolds number $\mathrm{Re}$ to the theoretically predicted value $\mathrm{Re}_{opt}$; connected white dots indicate the value of $\mathrm{Re}$ at which the minimal number iterations is achieved. The right column depicts the distribution of pressure and velocity streamlines. Panels **(a)**, **(b)**, and **(c)** correspond to problem configurations involving 1, 14, and 46 inclusions, respectively.





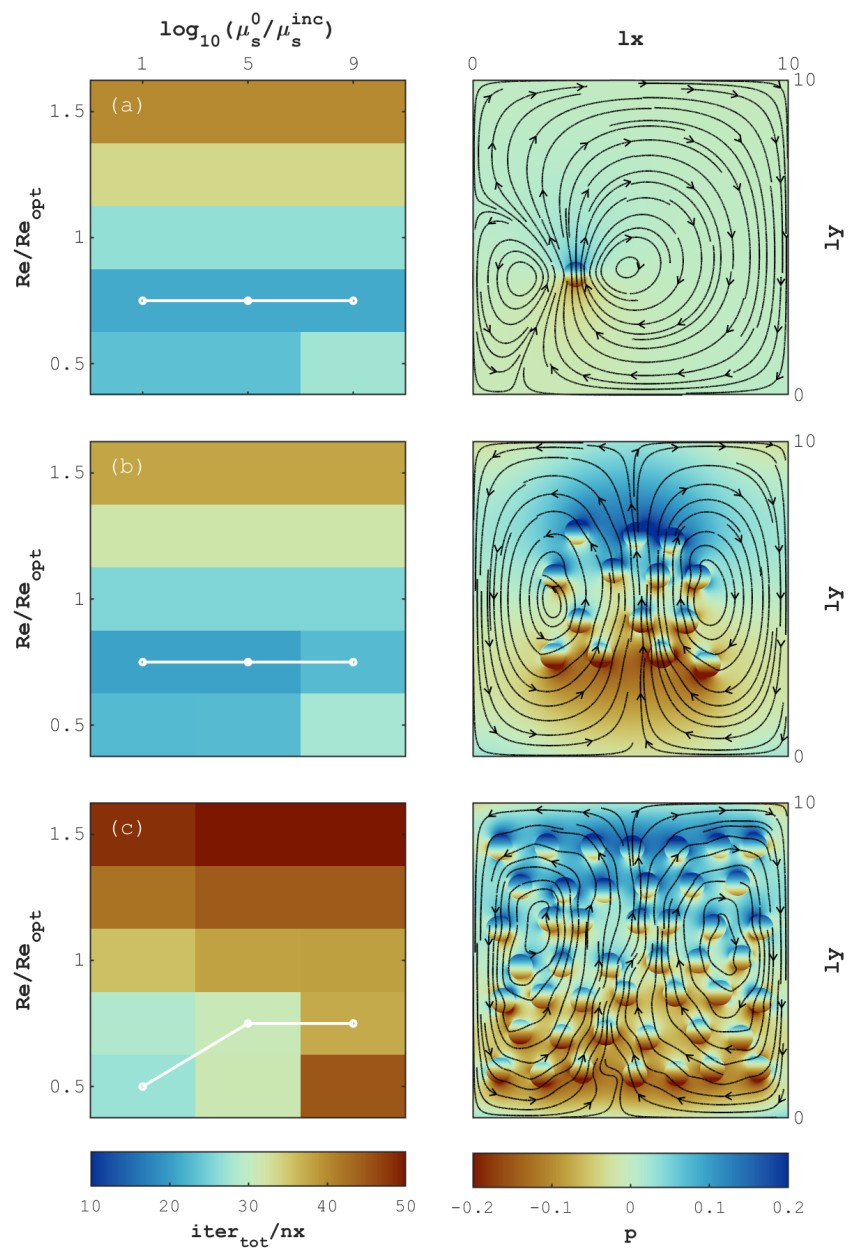

**Figure 13.** Buoyancy-driven flow. The left column report the number of iterations for different values of viscosity ratio $\mu_s^0/\mu_s^{inc}$ and ratio of numerical Reynolds number $\mathrm{Re}$ to the theoretically predicted value $\mathrm{Re_{opt}}$; connected white dots indicate the value of $\mathrm{Re}$ at which the minimal number iterations is achieved. The right column depicts the deviation of pressure from hydrostatic distribution and velocity streamlines. Panels **(a)**, **(b)**, and **(c)** correspond to problem configurations involving 1, 14, and 46 inclusions, respectively.





of iterations. The physically motivated iteration strategy enables one to control the stability of iterations through the single CFL-like parameter (see Sect. 2.1.2).

In both shear-driven and gravity-driven problem setups, the convergence is significantly slower than that for the single centred inclusion case. This slowdown could be explained by the complicated internal geometry involving non-symmetrical inclusion placement featuring huge viscosity contrasts which results in a stiff system.

## 8   Applications to nonlinear mechanical problems with elasto-viscoplastic rheology

To demonstrate the versatility of the approach, we tackle the nonlinear mechanical problem of strain localisation in two and
three dimensions. In the following applications we consider an elasto-viscoplastic (E-VP) rheological model, thus the serial viscous damper is deactivated and the flow is compressible. We assume a small-strain approximation. Hence, the deviatoric strain rate tensor may be decomposed in an additive manner in equation 42. A similar decomposition is assumed for the divergence of velocity in equation 43. The plastic model is based on consistent elasto-viscoplasticity and the yield function is defined as:

$$F = \tau_{\mathrm{II}} - p\sin\phi - c\cos\phi - \dot{\lambda}\mu^{\mathrm{vp}}, \tag{46}$$

where $\tau_{II}$ is the second stress invariant, $\phi$ is the friction angle, $\eta^{\mathrm{vp}}$ is the viscoplastic viscosity and $c$ is the cohesion. At the trial state, $F^{\mathrm{trial}}$ is evaluated assuming no plastic deformation ($\dot{\lambda} = 0$). Cohesion strain softening is applied and the rate of $c$ is expressed as:

$$\frac{\partial c}{\partial t} = \dot{\lambda}h, \tag{47}$$

where $h$ is a hardening/softening modulus. Viscoplastic flow is non-associated and the potential function is expressed as:

$$Q = \tau_{\mathrm{II}} - p\sin\psi, \tag{48}$$

where $\psi$ is the dilatancy angle. If $F \geq 0$, viscoplastic flow takes place and the rate of the plastic multiplier is positive and defined as:

$$\dot{\lambda} = \frac{F^{\mathrm{trial}}}{\mu^{\mathrm{ve}} + \mu^{\mathrm{vp}} + K\Delta t\sin\psi\sin\phi}. \tag{49}$$

The initial model configuration assumes a random initial cohesion field. Pure shear kinematics are imposed at the boundaries of the domain (see Sect. 5.2). The reader is referred to Duretz et al. (2019a) for the complete set of material parameters. We only slightly modify the value of $\mu^{\mathrm{vp}}$ with respect to Duretz et al. (2019a), such that $\mu^{\mathrm{vp}} = 9 \times 10^{18}$ Pa·s in the present study.

### 8.1   Performance benefits for desktop-scale computing

Besides the potential to tackle nonlinear multi-physics problem at supercomputer-scale, another critical aspect relies in the
ability to solve smaller-scale nonlinear problems. Here we investigate wall-times for the simulation of the previously-described



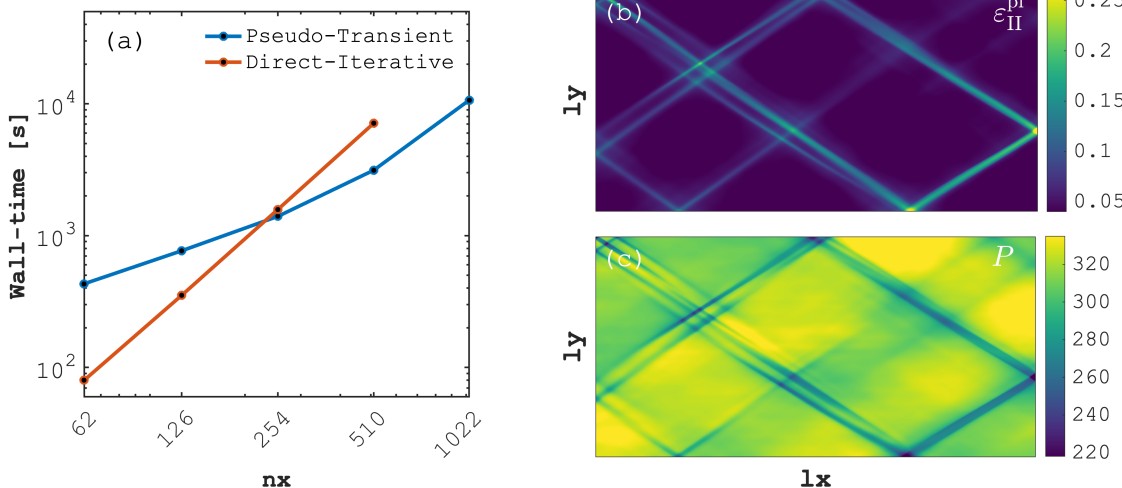

**Figure 14.** Performance comparison between the pseudo-transient and direct-iterative method resolving 2D shear band formation out of a random noise cohesion field. **(a)** Wall-time in seconds as function of numerical grid resolution in $x$-direction ($\sqrt{n_{\mathrm{dof}}}$). Panels **(b)** and **(c)** depict total accumulated plastic strain $\varepsilon_{\mathrm{II}}^{\mathrm{pl}}$ and pressure [MPa], respectively, from the $1022 \times 1022$ resolution 2D PT GPU simulation.

elasto-viscoplastic shear band formation in 2D. We compare a Matlab-based direct-iterative and a Julia GPU-based PT solver, respectively. The M2Di Matlab solver routines (Räss et al., 2017) rely on a Newton linearisation of the nonlinear mechanical problem (Duretz et al., 2019a) and use a combined direct-iterative (DI) solver to compute Newton steps. The solver is based on the combination of outer Powell-Hestenes and inner Krylov iterations (Global Conjugate Residual) that are preconditioned

with the Cholesky factorisation of the symmetrised Jacobian (Räss et al., 2017). In the 2D pseudo-transient solver, written in Julia using the ParallelStencil.jl packages, the evaluation of nonlinearities is embedded in the pseudo-time integration loop. The timings reported for the DI and the PT schemes were produced on a 2.9 GHz Intel Core i5 processor and on a single Nvidia Tesla V100 GPU, respectively. Each simulation resolves 100 physical (implicit) time steps. Models were run on resolutions involving $62^2$ to $510^2$ and up to $1022^2$ grid cells for the DI and PT scheme , respectively. As expected for 2D computations,

reported wall-times are lower using the DI scheme at very low resolutions. However, it is interesting to observe that the GPU-accelerated PT scheme can deliver comparable wall-times at already relatively low resolutions ($n_x \approx 254$). The employed CPU and GPU can be considered as common devices on current scientific desktop machine. We can thus conclude that the use of the GPU-accelerated PT schemes is a viable and practical approach to solve nonlinear mechanical problems on a desktop-scale computer. Moreover the proposed PT scheme turns out to be beneficial over common approaches (DI schemes) at relatively

low resolutions, already.



## 8.2 High-resolution 3D results

We here present preliminary 3D results of the spontaneous development of visco-plastic shear bands in pure shear deformation from an initial random cohesion field (Fig. 15). These 3D results further demonstrate the versatility of the pseudo-transient approach, enabling the seamless port of the 2D E-VP algorithm to 3D, extending recent work by (Duretz et al., 2019a) to
tackle three-dimensional configurations. We generate the 3D initial condition, a Gaussian random field with an exponential co-variance function, following the approach described in (Räss et al., 2019b), available through the ParallelRandomFields.jl Julia package (Räss and Omlin, 2021). We perform 100 implicit loading steps using the accelerated pseudo-transient method and execute the parallel Julia code relying on ParallelStencil.jl and ImplicitGlobalGrid.jl on 8 Tesla V100 on the *Volta* node.

Both the 2D and 3D elasto-viscoplastic algorithms require only minor modifications of the visco-elastic Stokes solver dis-
cussed throughout this manuscript to account for brittle failure, deactivation of the serial viscous damper and viscoplastic regularisation without significantly affecting the convergence rate provided by the second order method. These results support the robustness of the approach as predicting elasto-plastic deformation capturing brittle failure categorises as a rather "stiff" problem challenging the numerical solvers accordingly.

## 9    Discussion

The continuous development of many-core devices, GPUs at the forefront, increasingly shapes the current and future computing landscape. The fact that GPUs and latest multi-core CPUs turn classical workstations into personal supercomputers is exciting; tackling previously impossible numerical resolutions or multi-physics solutions becomes feasible as of technical progress. However, the current chip design challenges legacy serial and non-local or sparse matrix-based algorithms seeking at solutions to partial differential equations. Naturally, solution strategies designed to specifically target efficient large-scale computations
on supercomputers perform most efficiently on GPUs and recent multi-core CPUs, as the algorithms used are typically local and minimise memory accesses. Moreover, efficient strategies will not or only modestly rely on global communication and as a result exhibit close to optimal scaling.

We introduced the PT method in light of, mostly, iterative type of methods such as dynamic relaxation and semi-iterative algorithms (see, e.g., Saad, 2020, for additional details). These classical methods, as well as the here presented accelerated PT
method, implement "temporal" damping by considering higher-order derivatives with respect to pseudo-time. This contrasts with multi-grid or multi-level methods, building upon a "spatial" strategy based on space discretisation properties to damp the low-frequency error modes. Multi-grid, or multi-level methods are widely used to achieve numerical solutions in analogous settings as here described (Brandt, 1977; Zheng et al., 2013). Also, multi-grid methods may achieve convergence in $O(n_x)$ iterations upon employing optimal relaxation algorithm (Bakhvalov, 1966).

Besides their scalable design, most iterative methods are challenged by configurations including heterogeneities and large contrasts in material parameters, motivated by typical applications to a variety of geodynamics problems (e.g. Baumgardner, 1985; Tackley, 1996; May et al., 2015; Kaus et al., 2016). Well-tuned robust multi-grid solvers may overcome these limitations at the costs of more complex implementations. Our systematic investigation results (Sect. 5.2) suggest, however, the PT method



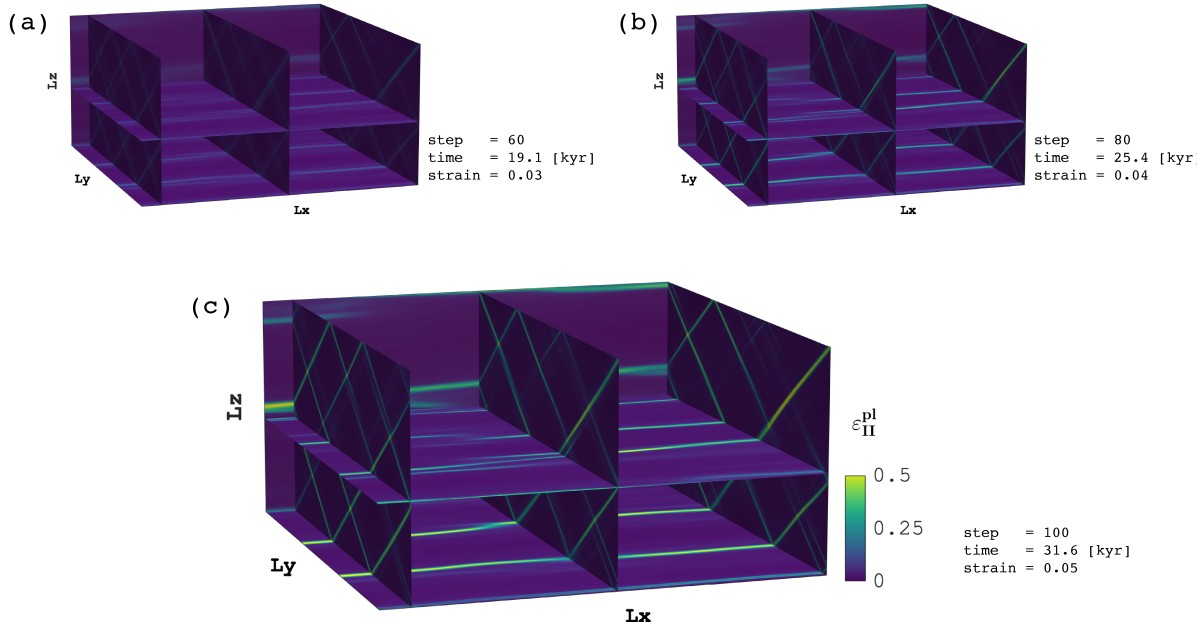

**Figure 15.** Total accumulated plastic strain $\varepsilon_{\mathrm{II}}^{\mathrm{pl}}$ distribution for a multi-GPU replica of the 2D calculation (Fig. 14b) resolving 3D shear band formation out of a random noise cohesion field. The numerical resolution includes $378 \times 378 \times 192$ grid cells. Panels depict elasto-plastic shear band formation after **(a)** 60 steps (corresponding 19.1 kyr and a strain of 0.03), **(b)** 80 steps (corresponding 25.4 kyr and a strain of 0.04), and **(c)** 100 steps (corresponding 31.6 kyr and a strain of 0.05).

to perform promisingly well, and at no specific additional design efforts w.r.t. the basic accelerated PT method implementation.
Beyond single phase mechanical considerations, the accelerated PT method delivers accurate solution in tightly coupled and nonlinear multi-physics problems (e.g. Duretz et al., 2019b; Räss et al., 2019a, 2020; Wang et al., 2021).

The ease of implementation lists among the main advantages of the accelerated PT method over other more complex ones such as, e.g., multi-grid. Particularly, all nonlinearities can be relaxed within a unique iteration loop, as reported in the non-linear diffusion results (Sect. 7.1). Often, due to the properties of the problem, the number of iterations does not exceed or is
even less than that in the case of constant coefficients. Other methods, in which nonlinear iterations are performed separately from the linear solver, cannot take advantage of the details of the physical process to reduce iteration counts. Also, for significantly nonlinear problems, e.g., associated with plastic strain localisation, thermo-mechanical shear-heating or two-phase fluid focusing, the physical processes occur on multiple spatial scales. Thus, ensuring an accurate description of these multi-scale solutions requires a high-resolution computational grid; it may be challenging for a coarse level multi-gird solve to provide
sufficient information in order to accurately resolve small-scale details only resolvable on the fine grid. Note that for analogous reasons, multi-grid methods are often used as a preconditioner for other iterative algorithms rather than the solution method.





Besides seeking at optimal convergence of the algorithm, the implementation efficiency also favours the accelerated PT method; the approach is simple but efficient, making it possible to further implement advanced optimisations such as explicit shared memory usage and register handling. The choice of a Cartesian regular grid allow for static and structures memory access patterns, resulting in access optimisation possibilities and balanced inter-process communications. Also, the absence of global reduction in the algorithm avoids severe bottlenecks. Finally, the amount of data transferred in the accelerated PT method is minimal, which allows achieving near-ideal scaling on distributed-memory systems, as reported in Sect. 7.3.2. Previous studies (e.g. Räss et al., 2018; Duretz et al., 2019b; Räss et al., 2019a, 2020) successfully implemented most of the algorithm designs here presented, although relying on a slightly different PT acceleration implementation (see Appendix B).

The main task in the design of PT methods is the optimal iteration parameters' estimation. For that, the spectral radius of the finite difference operator is often approximated based on the Gershgorin circle theorem (Papadrakakis, 1981). In this paper, we propose an alternative derivation of the PT algorithm entirely based on a physical analogy. The analysis of the convergence rate can be carried out within the framework of the spectral analysis of continuous equations, rather than the finite-dimensional space discretisation. The advantage of this approach relies in the availability of a physical interpretation of the iteration parameters, as well as in a clear separation of physics and numerics. For example, we show that for visco-elastic Stokes flow (Sect. 2.4), the pseudo-transient iteration parameters depend on the Reynolds number and the bulk-to-shear elastic modulus ratio. The stability of the iterative process is ensured by a choice of pseudo-physical material properties that is consistent with the conditions obtained on the basis of a von Neumann stability analysis.

The determination of the optimal iterative parameters is thereby reduced to the search for the optimal values of the dimensionless physical numbers that describe the properties of the underlying physical process. The addition of new physical processes, such as heat conduction, two-phase flow, chemical reactions, will lead to the natural emergence of new dimensionless parameters. Since many physical processes have a similar or even identical mathematical description, it is expected that the derivation of the accelerated PT method for such processes can be carried out similarly to those already developed. In this paper, such a derivation is provided for several important processes, namely, the linear and nonlinear diffusion, diffusion-reaction, non-stationary diffusion, and the visco-elastic Stokes problem. The efficiency of the accelerated PT method is demonstrated for essentially nonlinear problems, as well as for the problems with large contrasts in the material properties.

Recently, Wang et al. (2021) studied fluid flow in deformable porous media using an analogous numerical integration scheme. They show that under certain assumptions, the equations governing the dynamics of such two-phase flows reduce to a "double damped wave equation" system which is mathematically equivalent to the Eq. (12) and (6) describing the diffusion-reaction process (Sect. 2.1.3). They also report the optimal parameters obtained by dispersion analysis of these equations. These parameters are formulated by Wang et al. (2021) in terms of dimensional physical parameters. Through appropriate rescaling it is possible to recover the non-dimensional form presented in Sect. 2.1.3. We believe that our derivation in terms of non-dimensional variables helps to reveal the analogy between seemingly different physical properties and facilitates reusing the derived iteration parameters for various applications. We provide analysis for the variety of physical processes, including incompressible Stokes flow, in a unified manner, filling some of the gaps missing in previous studies.





**Table 1.** Wall-time prognostic for resolving the nonlinear diffusion and the visco-elastic Stokes 3D Julia multi-GPU applications on 2197 ($13^3$) Nvidia Tesla P100 GPUs on the *Piz Daint* supercomputer at CSCS.

| Application | GPU | $n_x$ local | $n_x$ global | $n_\mathrm{dof}^\mathrm{tot}$ | $t$/iter [s] | $n_\mathrm{iter}^\mathrm{tot}$ | wall-time |
|---|---|---|---|---|---|---|---|
| Nonlinear diffusion 3D | Tesla P100 | 512 | 6632 | $0.292 \times 10^{12}$ | 0.0348 | 12'932 | 7.5 min |
| Visco-elastic Stokes 3D | Tesla P100 | 383 | 4955 | $1.23 \times 10^{12}$ | 0.0644 | 408'788 | 7.31 hrs |

The scalability of the accelerated PT method as function of numerical resolution permits to predict the total iteration count, here for the nonlinear diffusion and the visco-elastic Stokes in 3D. The weak scaling benchmark results provide the time per iteration as function of the numerical resolution. Combining this information, it is possible to predict the time-to-solution or wall-time (Table 1) needed to resolve nonlinear diffusion and visco-elastic Stokes flow on $6632^3$ and $4955^3$ grid cells, respec-

tively, on 2197 Nvidia Tesla P100 GPUs on the *Piz Daint* supercomputer at CSCS. Single-GPU problem size scaling results for different GPU architectures further permit to extrapolate these wall-time to, e.g., the Nvidia Ampere or Volta architecture, as time per iteration is directly proportional to the effective memory throughput $T_\mathrm{eff}$. There is a $3.0\times$ and $1.5\times$ increase in $T_\mathrm{eff}$ on the A100 and V100 compared to the P100 architecture, respectively. resulting in wall-time in the order of 2.5 min and 2.4 hrs on a A100 powered system and of 5 min and 4.8 hrs a V100 powered system, for the nonlinear diffusion and the

visco-elastic Stokes solver, respectively.

In practical applications, the patterns of the flow may change drastically throughout the simulation owing to the spontaneous flow localisation or evolution of the interface between immiscible phases with significantly different properties. It is a requirement for the numerical method to be robust with respect to such changes. The iterative algorithm is expected to converge even in extreme cases, e.g., in the presence of sharp gradients across material properties, and the iteration parameters

should be insensitive to arbitrary changes in the internal geometry. We present a parametric study to assess the robustness of the accelerated PT method for typical building blocks for geophysical applications. We considered shear- and buoyancy-driven flows with multiple randomly distributed inclusions in a viscous matrix as proxies for more realistic problem formulations. We show that our method is capable of modelling flows with viscosity contrasts up to 9 orders of magnitude. The values of optimal iteration parameters obtained by the means of systematic simulation runs do not change significantly for a wide range

of material properties and internal configurations of the computational domain. We observe the significant slowdown in convergence for viscosity contrasts larger than 5 orders of magnitude in some of the considered cases. These results are expected given the ill-conditioned problem and motivate development of a scalable preconditioner suitable for massively parallel GPU workloads. The application of a robust preconditioner, with reference to previous discussion, may help to partly alleviate slow convergence. However, for viscosity contrasts of 6 orders of magnitude and above, a significant increase in the number of

iterations may be legitimate (May et al., 2015).

The numerical application to resolve shear-banding in elasto-viscoplastic media in 3D supports the versatility and the robustness of the presented approach putting emphasis on successfully handling complex rheology. These examples complement recent studies employing the accelerated pseudo-transient method to resolve spontaneous localisation owing to multi-physics





coupling (Räss et al., 2018; Räss et al., 2019a, 2020; Duretz et al., 2019b) and entire adjoint-based inversion frameworks
(Reuber et al., 2020). The Sect. B of the Appendix provides connections of the presented analysis with previous work.

## 10  Conclusions

The current HPC landscape redefines the rules governing applications' performance; the multi-core processors' massive paral-
lelism imposes a memory-bound situation. Our work shows that the most simple dynamic relaxation schemes, implementing
parabolic systems, turn out to report extremely low iteration count and to scale super-linearly with resolution increase when ele-
gantly transformed into hyperbolic systems using appropriate and physics-motivated pseudo-transient, or numerical, additions.
Moreover, the conciseness of the accelerated pseudo-transient approach permits the applications to execute at effective memory
throughput rate approaching memory copy rates (a theoretical upper bound) of latest GPUs. Further, hiding communication
behind computations permits to achieve parallel efficiency over 96% on various distributed memory systems and up to 2197
GPUs. The physics we selected for the numerical experiments represent key building blocks to further tackle various multi-
physics coupling, usually the combination of mechanical and diffusive processes. Our systematic results on the multi-inclusion
setup with huge viscosity contrasts provides some preliminary results assessing the robustness of the accelerated PT method,
which we further employ to resolve shear bad formation in 3D as a result of plastic yielding in elasto-viscoplastic materials.
Our study paves the way for resolving coupled and nonlinear multi-physics applications in natural sciences and engineering on
extremely high resolutions on next generation of exascale-capable supercomputers, revamping elegant iterative techniques and
implementing them with the portable Julia language.

*Code availability.* The various software developed and used in the scope of this study is licensed under MIT License. The latest ver-
sions of the code is available from GitHub at: https://github.com/PTsolvers/PseudoTransientDiffusion.jl and https://github.com/PTsolvers/
PseudoTransientStokes.jl (last access: 9 December 2021). Past and future versions of the software are available from a permanent DOI
repository (Zenodo) at: https://doi.org/10.5281/zenodo.5764691 (Räss and Utkin, 2021a) and https://doi.org/10.5281/zenodo.5764696 (Räss
and Utkin, 2021b). Both the PseudoTransientDiffusion.jl and the PseudoTransientStokes.jl repositories contain most Julia multi-XPU code
examples implementing the accelerated PT method and can be readily used to reproduce the figures of the paper. The codes are written using
the Julia programming language. Scaling figures can readily be reproduced using the available data, while all other figures require to first
execute the Julia scripts to produce the expected data. Refer to the repositories' `README` for additional information.

## Appendix A:  Dispersion analysis

### A1  First-order pseudo-transient diffusion

Let the total pseudo-time required to reach convergence be $\tau_{\mathrm{tot}}$. To estimate $\tau_{\mathrm{tot}}$ and the number of iterations in the numerical
implementation of the method, we consider the deviation $\epsilon$ of $H$ from the exact solution to Eq. (4). Linearity of the problem





makes it possible to reformulate the Eq. (5) in terms of $\epsilon$:

$$\frac{\partial \epsilon}{\partial \tau} = \nu \frac{\partial^2 \epsilon}{\partial x^2} , \tag{A1}$$

where $\nu = D/\tilde{\rho}$. Eq. (A1) is subject to homogeneous boundary conditions: $\epsilon(0, \tau) = \epsilon(L, \tau) = 0$.

We study the convergence rate of the pseudo-transient method by performing the dispersion analysis of Eq. (A1). The general solution to the Eq. (A1) with homogeneous Dirichlet boundary conditions has the following form:

$$\epsilon(x, \tau) = \sum_{k=1}^{\infty} E_k \exp\left(\frac{-\lambda_k V_d \tau}{L}\right) \sin\left(\frac{\pi k x}{L}\right) , \tag{A2}$$

where $V_d = \nu/L$ is the characteristic velocity scale of diffusion, $k$ is the wavenumber, $E_k$ and $\lambda_k$ are the initial amplitude and

the decay rate of the wave with wavenumber $k$, respectively. Only values of $k \geq 1$ are accounted for in Eq. (A2) because for other values of $k$ homogeneous boundary conditions are not satisfied.

We are interested in the exponential decay rate values $\lambda_k$, because they indicate the total time it takes for the error component with wavelength $\pi k$ to vanish. It is sufficient to consider only one typical term from the solution (A2):

$$\epsilon_k(x, \tau) = E_k \exp\left(\frac{-\lambda_k V_d \tau}{L}\right) \sin\left(\frac{\pi k x}{L}\right) . \tag{A3}$$

Substituting (A3) into Eq. (A1) gives

$$\lambda_k = \pi^2 k^2 . \tag{A4}$$

The dispersion relation (A4) indicates that the harmonics with higher wavenumber decay faster than the ones with smaller wavenumber. Therefore, the total time $\tau_{\text{tot}}$ to convergence is controlled by the component of $\epsilon$ with the smallest wavenumber $k = 1$. If the required non-dimensional accuracy, i.e. upper bound on $\epsilon$, is $\epsilon_{\text{tol}}$, then $\tau_{\text{tot}}$ can be obtained by solving Eq. (A3)

for $\tau$ at $k = 1$ and $x = L/2$:

$$\tau_{\text{tot}} = -\frac{L^2}{\pi^2 \nu} \ln\left(\frac{\epsilon_{\text{tol}}}{E_1}\right) . \tag{A5}$$

The stability analysis of Eq. (5) in discretised form suggests that the maximal time step $\Delta \tau$ allowed by the explicit integration scheme is proportional to the grid spacing $\Delta x$ squared: $\Delta \tau = \Delta x^2/\nu/2$. If the computational grid is uniform, as it is the case in this work, $L = n_x \Delta x$, where $n_x$ is the number of grid cells. Assuming $\tau_{\text{tot}} = n_{\text{it}} \Delta \tau$, where $n_{\text{it}}$ is the number of pseudo-

transient iterations required for convergence, and substituting expressions for $L$ and $\tau_{\text{tot}}$ into the Eq. (A5), we obtain:

$$n_{\text{it}} = -2 \frac{n_x^2}{\pi^2} \ln\left(\frac{\epsilon_{\text{tol}}}{E_1}\right) . \tag{A6}$$

Thus, the number of pseudo-time iterations required for convergence must be proportional to $n_x^2$.

## A2   Accelerated pseudo-transient diffusion

Similarly to the previous case, we reformulate the problem in terms of deviation from the exact solution and consider the

typical term from Fourier series expansion:

$$\epsilon_k(x, \tau) = E_k \exp\left(\frac{-\lambda_k V_p \tau}{L}\right) \sin\left(\frac{\pi k x}{L}\right) . \tag{A7}$$





Substituting (A7) into Eq. (7) gives the following dispersion relation:

$$\lambda_k^2 - \mathrm{Re}\lambda_k + \pi^2 k^2 = 0 \ , \tag{A8}$$

where $\mathrm{Re} = LV_{\mathrm{p}}/\nu = \sqrt{L^2/\nu/\theta_{\mathrm{r}}}$. The non-dimensional parameter $\mathrm{Re}$ can be interpreted as a numerical Reynolds number, or
as the inverse of a numerical Deborah number's square root, characterising the ratio between relaxation time $\theta_{\mathrm{r}}$ and characteristic time of diffusion $L^2/\nu$.

Solving the Eq. (A8) for $\lambda_k$ yields:

$$\lambda_{k_{1,2}} = \frac{\mathrm{Re} \pm \sqrt{\mathrm{Re}^2 - 4\pi^2 k^2}}{2} \ . \tag{A9}$$

Depending on values of $\mathrm{Re}$ and $k$, $\lambda_k$ can be complex. The imaginary part of $\lambda_k$, $\Im(\lambda_k)$, contributes to the oscillatory behaviour
of the solution, while the real part $\Re(\lambda_k)$ controls the exponential decay of deviation $\epsilon$. The minimum of $\Re(\lambda_k)$ between the two roots of Eq. (A9) indicates the decay rate of the entire solution. This minimum reaches maximal value when both roots of Eq. (A9) are equal, i.e., when the discriminant of (A9) $D = \mathrm{Re}^2 - 4\pi^2 k^2 = 0$. Therefore, $\mathrm{Re} = \pm 2\pi k$, but negative $\mathrm{Re}$ leads to exponential growth of the $\exp(-\lambda_k V_p t/L)$ term in the solution (A7), which is non-physical. Finally, choosing iteration parameters for each wavelength independently is not realistic, we chose $\mathrm{Re}$ such that the longest wave with wavenumber $k = 0$
is damped most effectively. The optimal value of $\mathrm{Re}$ is then given by

$$\mathrm{Re}_{\mathrm{opt}} = 2\pi \ . \tag{A10}$$

Total pseudo-time $\tau_{\mathrm{tot}}$ required for convergence with tolerance $\epsilon_{\mathrm{tot}}$ is determined in the same way as in the previous case:

$$\tau_{\mathrm{tot}} = \frac{L}{\pi V_p} \ln\left(\frac{E_1}{\epsilon_{\mathrm{tol}}}\right) \tag{A11}$$

The stability analysis of the damped wave equation in discretised form suggests that in a certain range of values of $\theta_{\mathrm{r}}$, the
maximal time step $\Delta\tau = \Delta x/V_{\mathrm{p}}$ (Alkhimenkov et al., 2021a). The number of pseudo-time iterations required for convergence $n_{\mathrm{it}}$ can then be estimated as

$$n_{\mathrm{it}} = \frac{n_x}{\pi} \ln\left(\frac{E_1}{\epsilon_{\mathrm{tol}}}\right) \ , \tag{A12}$$

and proportional to $n_x$ in the accelerated scheme.

## A3   Diffusion-reaction

Reformulating Eq. (13) in terms of the difference $\epsilon$ between the exact steady-state solution and the pseudo-transient solution and substituting it into Eq. (A7), one obtains the following dispersion relation:

$$\lambda_k^2 - \left(\frac{\mathrm{Da}}{\mathrm{Re}} + \mathrm{Re}\right)\lambda_k + \mathrm{Da} + \pi^2 k^2 = 0 \ . \tag{A13}$$





Introducing the auxiliary parameter $\widehat{\mathrm{Re}} = \mathrm{Da}/\mathrm{Re} + \mathrm{Re}$, it can be seen that the dispersion relation (A13) is now similar to the one reported in Eq. (A8). The solution for $\lambda_k$ is now given by

$$\lambda_{k_{1,2}} = \frac{\widehat{\mathrm{Re}} \pm \sqrt{\widehat{\mathrm{Re}}^2 - 4\pi^2 k^2 - 4\mathrm{Da}}}{2} \; . \tag{A14}$$

The resulting optimal value is $\widehat{\mathrm{Re}} = 2\sqrt{\pi^2 + \mathrm{Da}}$, and the optimal value for $\mathrm{Re}$ is obtained by solving $\widehat{\mathrm{Re}}(\mathrm{Re})$ for $\mathrm{Re}$:

$$\mathrm{Re}_{\mathrm{opt}} = \pi + \sqrt{\pi^2 + \mathrm{Da}} \tag{A15}$$

As expected, in the limit of $\mathrm{Da} \to 0$, i.e., when the process becomes diffusion-dominated, the optimal value of the parameter $\mathrm{Re}$, determined by Eq. (A15), coincides with the value given by Eq. (A10) in the case of a purely diffusive process.

The number of iterations required to converge to a tolerance $\epsilon_{\mathrm{tol}}$ is

$$n_{\mathrm{it}} = \frac{n_x}{\sqrt{\pi^2 + \mathrm{Da}}} \ln\left(\frac{E_1}{\epsilon_{\mathrm{tol}}}\right) \; . \tag{A16}$$

Interestingly, the iteration count estimate given by Eq. (A16) decreases proportionally to $\sqrt{\mathrm{Da}}$. Therefore, the number of iterations for a diffusion-reaction process will always be lower than for a pure diffusion process if $\mathrm{Da} > 0$.

## A4   Incompressible viscous Stokes

The system of equations (25)–(27) can be reduced to a single equation for velocity $v_x$:

$$\mu_s \frac{\tilde{\rho}}{\widetilde{G}} \frac{\partial^3 v_x}{\partial \tau^3} - \mu_s \left(\frac{\widetilde{K}}{\widetilde{G}} + 2\right) \frac{\partial^3 v_x}{\partial \tau \partial x^2} + \tilde{\rho} \frac{\partial^2 v_x}{\partial \tau^2} = \widetilde{K} \frac{\partial^2 v_x}{\partial x^2} \; . \tag{A17}$$

Following the established procedure, we reformulate Eq. (A17) in terms of deviation $\epsilon$ from the exact solution, and consider typical terms in Fourier series expansion of $\epsilon$:

$$\epsilon_k(x, \tau) = E_k \exp\left(\frac{-\lambda_k V_p \tau}{L}\right) \sin\left(\frac{\pi k x}{L}\right) \; . \tag{A18}$$

Substituting Eq. (A18) into Eq. (A17), we obtain the following dispersion relation:

$$\lambda_k^3 - \mathrm{Re}\lambda_k^2 + \pi^2 k^2 (r+1)\lambda_k - \pi^2 k^2 r \mathrm{Re} = 0 \; . \tag{A19}$$

Depending on the values of the coefficients, the dispersion relation (A19) would have either one real root and two complex conjugate roots or three real roots.

For analysis, it is useful to recast the dispersion relation (A19) in depressed form $x^3 + px + q$ applying a change of variables $\alpha_k = \lambda_k - \mathrm{Re}/(3(r+2))$:

$$\alpha_k^3 + \left[\pi^2 k^2 - \frac{\mathrm{Re}^2}{3(r+2)^2}\right]\alpha_k +$$

$$\frac{2\mathrm{Re}}{9(r+2)}\left[\pi^2 k^2 \left(1 + \frac{1-4r}{r+2}\right) - \frac{\mathrm{Re}^2}{3(r+2)^2}\right] = 0 \; . \tag{A20}$$



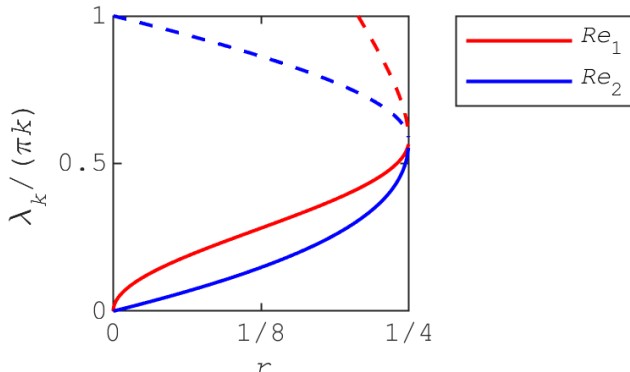

**Figure A1.** Roots of the dispersion relation $\lambda_k$ for two branches of the numerical parameter Re. Solid and dashed lines show maximal and minimal roots, respectively.

The discriminant of the depressed cubic equation is $x^3 + px + q$ is $-4p^3 - 27q^2$. Setting the discriminant of Eq. (A20) to zero yields:

$$r\mathrm{Re}^4 - (1 - 2r^2 + 10r)(r+2)\pi^2 k^2 \mathrm{Re}^2 + (r+2)^5 \pi^4 k^4 = 0 . \tag{A21}$$

Eq. (A21) is biquadratic w.r.t. Re. We denote the discriminant of Eq. (A21) by $D$. For Re to be real, $D$ must be non-negative:

$$D = \pi^4 k^4 (r+2)^2 (1 - 4r)^3 \geq 0 \iff r \leq \frac{1}{4} . \tag{A22}$$

By definition, $r$ is the ratio between the bulk and shear modulus, therefore, $r$ must be positive. Thus, $r \in (0; 1/4]$.

By solving Eq. (A21), we obtain the relation between Re and $r$:

$$905 \quad \mathrm{Re}_{1,2} = \sqrt{\frac{(r+2)\left(1 - 2r^2 + 10r \pm \sqrt{(1-4r)^3}\right)}{2r}} \pi k . \tag{A23}$$

Only the positive branch of the solution is taken in (A23) because Re must be positive.

When the depressed cubic equation $x^3 + px + q$ has multiple roots, the simple root is $x_1 = 3q/p$, and the double root is $x_2 = x_3 = -3q/(2p)$. Substituting expressions for $q$ and $p$ from the dispersion relation (A20), and changing variables back to $\lambda_k$ from $\alpha_k$, one obtains:

$$910 \quad \lambda_{k_1} = \frac{\mathrm{Re}}{r+2}\left(1 + \frac{2(r+2)(1-4r)\pi^2 k^2}{3(r+2)^2 \pi^2 k^2 - \mathrm{Re}^2}\right) , \tag{A24}$$

$$\lambda_{k_{2,3}} = -\mathrm{Re}\frac{(1-4r)\pi^2 k^2}{3(r+2)^2 \pi^2 k^2 - \mathrm{Re}^2} . \tag{A25}$$

Substituting Eq, (A23) into Eq. (A24), one obtains roots of the dispersion relation (A19) as a function of parameter $r$. The roots depend monotonously on $r$ for all choices of Re (Fig. A1). The minimal root reaches its maximum value at $r_{\mathrm{opt}} = 1/4$. Corresponding value of Re is then $\mathrm{Re}_{\mathrm{opt}} = 9\sqrt{3}\pi/4$ for $k = 1$.



**Appendix B:  Connection to the previous work**

It is useful to provide an analogy between the presented analysis and some previous studies, namely, pseudo-transient continuation model by (Räss et al., 2018; Räss et al., 2019a, 2020; Duretz et al., 2019b), and early work by (Frankel, 1950). For this comparison, we consider only stationary diffusion process with $D = \mathrm{const}$.

We reformulate the damped wave equation, Eq. (7), as a first-order system introducing $R$, the pseudo-time derivative of $H$:

$$\frac{\partial H}{\partial \tau} = R \ , \tag{B1}$$

$$\theta_{\mathrm{r}} \frac{\partial R}{\partial \tau} + R = \frac{1}{\tilde{\rho}} \nabla_k D \nabla_k H \ . \tag{B2}$$

In all mentioned studies the numerical discretisation of Eq. (B1) and (B2) were considered. Let $f^k$ be the finite-difference operator approximating the right-hand side of Eq. (B2) at time step $k$. Using a forward Euler scheme for the time integration, one obtains:

$$\frac{H_i^k - H_i^{k-1}}{\Delta \tau} = R_i^k \ , \tag{B3}$$

$$\theta_{\mathrm{r}} \frac{R_i^k - R_i^{k-1}}{\Delta \tau} + R_i^{k-1} = f^k \ . \tag{B4}$$

Let $g_i^k = \theta_{\mathrm{r}} R_i^k / \Delta \tau$. Rearranging Eq. (B3) and (B4) to formulate the update rules for $H_i^k$ and $g_i^k$:

$$H_i^k = H_i^{k-1} + \frac{\Delta \tau^2}{\theta_{\mathrm{r}}} g_i^k \ , \tag{B5}$$

$$g_i^k = f_i^k + \left(1 - \frac{\Delta \tau}{\theta_{\mathrm{r}}}\right) g_i^{k-1} \ , \tag{B6}$$

and using the definitions of $\theta_{\mathrm{r}}$ and $\Delta \tau$ reported by Eq. (10) and (8):

$$\frac{\Delta \tau}{\theta_{\mathrm{r}}} = \widetilde{C} \frac{\mathrm{Re} L}{\Delta x} = \widetilde{C} \frac{\mathrm{Re}}{n_x} \ , \tag{B7}$$

it is evident that, if combined, Eq. (B5), (B6) and (B7) are equivalent to the formulation of pseudo-transient continuation method presented in Räss et al. (2018). Substituting Eq. (B6) into Eq. (B5) and expressing $g_i^{k-1}$ in terms of $H_i^{k-1}$ and $H_i^{k-2}$ yields

$$H_i^k = H_i^{k-1} + \frac{\Delta \tau^2}{\theta_{\mathrm{r}}} f^k + \left(1 - \frac{\Delta \tau}{\theta_{\mathrm{r}}}\right) \left(H_i^{k-1} - H_i^{k-2}\right) \ , \tag{B8}$$

which is equivalent to the second-order Richardson method proposed in Frankel (1950).

*Author contributions.*  **LR** Original study design, code and algorithm development, Julia at scale implementations, scaling and performance benchmarks, figure creation, manuscript edition. **IU** Code and algorithm development, dispersion analysis, parametric study, figure creation,





manuscript edition. **TD** Original study design, low-performance computations, visco-elasto-plastic model investigations, prototyping, figure

design, manuscript edition. **SO** Building block development for Julia at scale implementations, effective throughput metric development, Julia deployment on Piz Daint, early and ongoing work for scalable architecture-agnostic PDE solver implementation, manuscript edition. **YP** Early work on accelerated PT method, dispersion analysis, methodology and PDE solver implementation on GPU.

*Competing interests.*  The authors declare that they have no conflict of interest.

*Acknowledgements.*  We thank Boris Kaus, Stefan Schmalholz and Mauro Werder for stimulating discussions. We acknowledge the Swiss

Geocomputing Centre, University of Lausanne, for computing resources on the *Octopus* supercomputer and are grateful to Philippe Logean for continuous technical support. **LR**, **IU** and **SO** acknowledge financial support from the Swiss University Conference and the Swiss Council of Federal Institutes of Technology through the Platform for Advanced Scientific Computing (PASC) program. This work was supported by a grant from the Swiss National Supercomputing Centre (CSCS) under project ID c23, obtained via the PASC project GPU4GEO. **IU** and **YP** acknowledge the financial support from the Russian Ministry of Science and Higher Education (project No. 075-15-2019-1890).





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
