# Peer review of "Assessing the robustness and scalability of the accelerated pseudo-transient method"

_Geoscientific Model Development, 2021_

## Author Response (AR1)

Geosci. Model Dev. Discuss., author comment AC1
https://doi.org/10.5194/gmd-2021-411-AC1, 2022

[Figure]

**Reply on RC1**

Ludovic Räss et al.
* * *
Author comment on "Assessing the robustness and scalability of the accelerated pseudo-transient method towards exascale computing" by Ludovic Räss et al., Geosci. Model Dev. Discuss., https://doi.org/10.5194/gmd-2021-411-AC1, 2022
* * *
Dear Dr. Wang,

Thank you for reviewing our mansucript draft. We addressed all your suggestions and hope we provided answers to the still open questions you pointed out. Find hereafter a detailed reply to your comments.

Best regards,

Ludovic Räss on behalf of the authors.

Detailed replies:

**Line 8: 1.2 trillion degrees of freedom**
**I noticed 1.2trillion is not equal but about 10 time larger than 4995^3, which cause my confusion. This means there would be about 10 different physical variables in each cell. But I don't think it is mentioned anywhere directly in manuscript.**

Thanks for reporting this. It was indeed not very clear. We now provide a new table listing the number of DoFs per grid point as well as the number of fields used to compute the $T_{eff}$ metric, and precised how to compute the total number of DoFs we report.

**Line 12: low resolution**
**It would be nice to mention how low the resolution, like 254x254?**

Thanks for suggesting. The resolution is given in the main text and we feel this information may not significantly enhance the abstract, thus we prefer not to include it.

**Line 90: wave-like or mechanical process**
**Why would mechanical processes be the same with wave like process? It needs more explanation if it is written like this.**

We do not state wave-like processes being the same as mechanical ones. We say that many geo-processes of interest can fall in one of these three categories.

**Line 135: "The choice of the boundary conditions type affects only the values of optimal iteration parameters"**
**Is there example in this study? Or do you mean boundary condition affect the iteration count?**

We mean what is written in the text, i.e., changing the type of boundary conditions will have an impact on the optimal values of the iteration parameters.

**Line 212: "the iteration parameters"**
**It is probably better to specify which parameter should be locally defined. "C" is also iteration parameter. Do you change it locally?**

We now explicitly enumerate the iteration parameters.

**L. 484: single-loop iterative procedure,**
**I found this sentence is a bit confusing. You have dual time iteration: inner loop and outer loop. Here you say single-loop.**

Thanks for pointing this out. We clarified the situation in the text. We here have one single iterative loop (combining the nonlinear and linear solve that usually require two nested loops). On top of this implicit iterative solver, we have the physical time loop.

**Fig 5 and line 538-540 and line 546-547**
**It show that 3D case (yellow line) require higher value of normalized iteration count. This is just the opposite with what line 538 says. Explain?**

Thanks for highlighting this poor explanation. We rephrased the sentence making the example clearer.

**Line 560-561. 17 nx for 1023x1023 is good**

It is good in the context of this study.

**Line 590. This sentence for a single paragraph? Fig 9 caption has already said something about this. Perhaps this sentence can be removed.**

Thanks, we added the missing info to the caption and removed the sentence.

**Line 605-607. What does "the best known single-XPU implementation" refer to. I can not see from the context. A bit confusing for me. Would "the parallel efficiency of a single GPU is also below 100%" sound better?**
**It should be "than" instead of "then" in line 607. I also notice there are other place "then" is used instead of "than". Please check!**

We sightly re-phrased the paragraph. Explanations in the follow sentence should be sufficient to set the context.

We also fixed the then vs than in the entire document.

**Fig 11. What might be the reason for Tesla A100 behave differently in the diffusion and stokes solver? It was the worse parallel efficiency for the diffusion problem and it become the best for stokes problem.**

Thanks, we added some suggestions in the text.

**Line 630-631 and Fig 12**
**This description is not consistent with Fig 12. Please check! Also, why would viscosity contrast of 1e5 need higher iteration time than viscosity contrast of 1e6-1e9 in F12.b,c.?**

The pseudo-transient method convergence rate is defined in the robustness study by the interaction between the "internal geometry", i.e., the spatial distribution of the material properties, and the boundary conditions. In the general case, the accurate prediction for the iteration count is possible only through the proper spectral analysis of the discrete finite-difference operator. Such an analysis could be impractical for large-scale problems. We aimed to demonstrate in that study that the simple analytical estimate for the optimal iteration parameter Re that we present in the paper remains a good starting point for numerical experiments that gives satisfactory convergence rate.

**Line 634-635**
**Which iteration parameter do you use local values in each grid cell? Re?**

We clarified the parameters and added a cross ref.

**Line 651. Eta_vp is not consistent with Eq.46.**

Thanks, we fixed it.

**Line 794: extremely low**
**I agree it is very low. But it would be nice to have a comparison when one say "extremely low". What are the normal/standard value for the iteration count when other iterative method is used!**

We acknowledge the lack of clarity and rephrased the sentence.

**Line 795. Or numerical additions**
**It is not clear here what you want to express here!**

We acknowledge the lack of clarity and rephrased the sentence.

**Line 802: shear bad**
**You mean "shear band", I suppose.**

Thanks for spotting that one.

[Figure]

Geosci. Model Dev. Discuss., author comment AC2
https://doi.org/10.5194/gmd-2021-411-AC2, 2022

[Figure]

**Reply on RC2**

Ludovic Räss et al.
* * *
Author comment on "Assessing the robustness and scalability of the accelerated pseudo-transient method towards exascale computing" by Ludovic Räss et al., Geosci. Model Dev. Discuss., https://doi.org/10.5194/gmd-2021-411-AC2, 2022
* * *
Dear reviewer,

We thank you for the suggestions made to our initial mansucript draft and addressed your major and minor comments. Please find hereafter the answers to your comments and suggestions.

Thank you for your insights

Ludovic Räss on behalf of the authors

Detailled replies:

**Title: "towards exascale computing" is not necessary. Remove.**

We removed it.

**Major Comments:**
**1. In the introduction, the authors contrast the pseudo-transient methods with Krylov iteration methods, such as conjugate gradient or GMRES methods. A benefit of pseudo-transient is that they are local and do not require global reductions unlike standard Krylov methods. First, there has been work on communication avoiding Krylov methods that reduces/avoids many of these global comms. See, for example, the widely cited Ph.D. thesis:**
**Hoemmen, Mark. Communication-avoiding Krylov subspace methods. University of California, Berkeley, 2010, or the more recent work the reduces the number global reductions for Gram-Schmidt and GMRES:**
**Åwirydowicz, Katarzyna, et al. "Low synchronization Gram−Schmidt and generalized minimal residual algorithms." Numerical Linear Algebra with Applications 28.2 (2021): e2343.**

Thank you providing these references. We added one paragraph to the introduction including and discussing them.

**In addition, preconditioning and "intelligent" guesses for the initial Krylov vector can vastly reduce the number of iterations required, thus making Krylov methods more competitive. A computational comparison and discussion of the proposed method with Krylov would be a welcome addition to the paper.**

A comparison between advanced communication-hiding Krylov solvers and the accelerated PT method would certainly be valuable. However, this initiative represents a project on its own and goes beyond the scope of this study. This will be addressed in a subsequent study.

**2. In Section 2, the authors assume the the computational domain is a cube with the same number of cells in each dimension. In geoscientific models, such as the atmosphere and the ocean, there is are order of magnitude differences in scales between the horizontal and vertical, and hence large differences in the grid spacing. The PT methods requires choosing an optimal Reynolds number, which depends on the length scale. How would the authors adapt the PT method to handle these scale differences--they claim "the solution strategy is not restricted to cubic meshes with similar resolution..."**

Good point. We added one more figure (Fig. 14) reporting the normalised iteration count for various numerical resolutions varying the aspect ratio from 1 to 8 using the visco-elastic Stokes flow in 2D. We show that the convergence is not hindered by larger aspect ratio while keeping the cell aspect ratio constant.

**3. The English is sub-standard and needs to be improved. See the minor comments.**

We worked on it.

**Minor Comments/questions:**
**1. Line 31: "see a regain in active development..." is awkward. Replace with "are in active development". Citations to back this assertion would be nice.**

We removed "active" but did not further change the sentence as the suggested changes do no longer convey our message, namely, "there is a regain in development" is not similar to "there is an active development".

**2. Line 121: The notation [0;L] is not standard. [0,L] is standard.**

We've changed it.

**3. Equation (1): Odd notation for the divergence operator on the right hand side. This is the continuity equation assuming constant density.**

Thank you for the comment. Note, however, that we never refer throughout the manuscript about Equation (1) standing for diffusion of mass (for which case some minor inconsistencies with respect to density may rise). Within this study we use the diffusion equation as inspired by, e.g., heat diffusion. The here diffusion (and not continuity) equation has the rho parameter as a factor of proportionality written down as such for internal consistency with the following steps.

**4. Equation (2): Why is "i" used instead of "k" ?**

K is for summation, i is per dimension. We added some clarification to the text.

**5. Line 138: Replace "to assemble" with "assembling"**

We fixed it.

**6. Line 146: Replace Eq. (15) with Eq. (3).**

We fixed it.

**7. Line 149: Replace the comma after tau with a semicolon.**

We fixed it.

**8. Line 174: Remove "the" before "Eq. (7)".**

We fixed it.

**9. Equation 7: This is the equation for Cattaneo diffusion. See, for example, "Methods of Theoretical Physics" by Morse and Feschbach. It is also called the Telegrapher's Equation in .**

Thank you for the comment. We did not modify the text since the suggested naming may not be that standard.

**10. Line 181: What happens to C if the grid spacing is different in different dimensions? See major comment 2.**

One could expect some loss of accuracy in the FD scheme.

**11. Line 335: Replace "it's" with "its".**

We fixed it.

---

## Author Response (AR2)

Dear Dr. Kelly,

Please find enclosed the latest version of our manuscript draft including the minor changes to Fig. 15 as requested by Dr. Kaus. We believe these clarifications will enhance the readability and relevance of the results.

We improved the GPU data and re-designed the wall-time plot to better reflect the advantages of using the PT GPU-based solver.

We wish the manuscript in the current form to be receivable and thank you for handling our submission.

Best regards,

Ludovic Räss, on behalf of the authors